# Rel-HNN: Split Parallel Hypergraph Neural Network for Learning on Relational Databases

**Md. Tanvir Alam**                                                    *tanvir15@du.ac.bd*
*Department of Computer Science and Engineering*
*University of Dhaka*

**Md. Ahasanul Alam**                                                  *ahasan@cse.du.ac.bd*
*Department of Computer Science and Engineering*
*University of Dhaka*

**Md Mahmudur Rahman**                                             *mahmudur@cse.du.ac.bd*
*Department of Computer Science and Engineering*
*University of Dhaka*

**Md Mosaddek Khan**                                                *mosaddek@du.ac.bd*
*Department of Computer Science and Engineering*
*University of Dhaka*

**Reviewed on OpenReview:** *https://openreview.net/forum?id=L7VP7gxpVG*

## Abstract

Relational databases (RDBs) are ubiquitous in enterprise and real-world applications. Flattening the database poses challenges for deep learning models that rely on fixed-size input representations to capture relational semantics from the structured nature of relational data. Graph neural networks (GNNs) have been proposed to address this, but they often oversimplify relational structures by modeling all the tuples as monolithic nodes and ignoring intra-tuple associations. In this work, we propose a novel hypergraph-based framework, that we call rel-HNN, which models each unique attribute-value pair as a node and each tuple as a hyperedge, enabling the capture of fine-grained intra-tuple relationships. Our approach learns explicit multi-level representations across attribute-value, tuple, and table levels. To address the scalability challenges posed by large RDBs, we further introduce a split-parallel training algorithm that leverages multi-GPU execution for efficient hypergraph learning. Extensive experiments on real-world and benchmark datasets demonstrate that rel-HNN significantly outperforms existing methods in both classification and regression tasks. Moreover, although the benefits of split-parallel training diminish on smaller hypergraphs with fewer nodes due to communication overhead, it achieves substantial speedups of up to 3.18× on large-scale relational datasets and up to 2.94× on large hypergraph datasets.

## 1    Introduction

Relational databases (RDBs) are among the most widely used forms of data representation in enterprise environments, owing to their ability to efficiently handle structured data, support complex queries, and maintain data integrity. A relational database consists of multiple tables governed by a schema that defines the relationships among them. RDBs serve as the primary data format across a wide range of industries, including online advertising, recommender systems, healthcare, and fraud detection. Despite their widespread use, the direct application of machine learning—particularly deep learning—to relational databases (RDBs) has received limited attention from the research community. Traditionally, applying machine learning to relational databases requires transforming the data into a flat, tabular format, since most supervised learning

models rely on fixed-size input vectors. This transformation process, commonly referred to as *flattening*, typically involves joining multiple related tables into a single denormalized table with predefined columns. Flattening usually demands extensive, rule-based feature engineering (Anderson et al., 2013; Bahnsen et al., 2016; Covington et al., 2016), and frequently results in the loss of valuable relational information embedded within the schema and data. Moreover, for large-scale RDBs, flattening introduces substantial computational overhead, becoming a major bottleneck in the overall machine learning pipeline. Consequently, there is an increasing demand for machine learning approaches capable of directly operating on relational data in its native form, without the need for manual feature engineering or flattening.

Effective application of neural network-based techniques to relational databases hinges on overcoming two fundamental challenges. First, the inherently structured yet complex nature of relational databases, characterized by multiple interconnected tables, demands models capable of capturing and leveraging these intricate relationships. Second, the considerable scale of relational databases—often containing millions or even billions of records across dozens to hundreds of tables—necessitates efficient training and inference procedures to ensure both practicality and scalability (Hilprecht et al., 2023). A recent approach that has gained traction to address these challenges involves applying Graph Neural Networks (GNNs) directly to relational databases (Li et al., 2016; Hamilton et al., 2017; Kipf & Welling, 2017; Velickovic et al., 2018; Xu et al., 2019; Cvitkovic, 2020b). In this context, the primary task typically involves predicting values for a target column in a specific table, using available relational information from the entire database. To achieve this, databases are modeled as graphs, where tuples serve as vertices and relationships between tuples from different tables—defined by foreign keys—serve as edges. To better capture relational semantics, existing methods have proposed utilizing relation-type-dependent weights (Schlichtkrull et al., 2018), specialized convolution operators (Huang et al., 2020), or generative architectures (Sun et al., 2019) within relational graphs.

Despite their expressive power, existing GNN-based approaches have notable limitations when modeling relational databases. Primarily, they treat entire tuples as monolithic nodes, thus ignoring the granular, attribute-level structures and failing to capture fine-grained inter-attribute interactions. Moreover, relying solely on primary key–foreign key (PK–FK) relationships significantly limits their ability to represent meaningful relationships among tuples within the same table. Additionally, these methods often overlook symmetries among sub-graphs and require multiple rounds of message passing, resulting in inefficient training and inference processes, particularly at scale. Furthermore, existing methods are heavily schema-dependent, making them challenging to generalize or adapt to new datasets without substantial human effort to define or extract PK–FK constraints explicitly.

An emerging alternative is hypergraph-based modeling, which has recently shown promise for automated learning on relational databases (Bai et al., 2021). A hypergraph generalizes traditional graphs by allowing edges (termed hyperedges) to connect an arbitrary number of vertices rather than being restricted to pairs. A hypergraph can reveal complex structural patterns involving multiple vertices and provide insights into network dynamics—patterns that traditional graphs, limited to pairwise connections, fail to capture (Kim et al., 2024). This ability to represent higher-order interactions among vertices has gained significant attention across various real-world complex systems, including physical systems (Battiston & Petri, 2022), microbial communities (Morin et al., 2022), brain functions (Expert & Petri, 2022), and social networks (Iacopini et al., 2022). The flexibility of hypergraphs in modeling multi-way interactions has motivated the development of powerful hypergraph neural network (HGNN) algorithms (Feng et al., 2019; Chien et al., 2022; Yadati et al., 2019), facilitating the learning of intricate relational patterns.

For relational databases specifically, ATJ-Net (Bai et al., 2021) leverages a hypergraph to train a heterogeneous GNN. It initially represents joinable attributes as vertices and tuples as hyperedges. The hypergraph is then transformed into to a heterogenous bipartite graph where the tuples and joinable attributes constitute vertices. Then, it applies a message-passing GNN to the bipartite graph to predict labels associated with tuples in the target table. Although ATJ-Net includes joinable attributes alongside tuples, it has several critical limitations. First, it considers only categorical, joinable attributes—typically those defined explicitly through primary key–foreign key (PK–FK) relationships—thereby confining the model to predefined relational paths and potentially missing complex attribute associations within tuples. Second, transforming the hypergraph into a bipartite graph inevitably flattens high-order relationships into pairwise edges, causing the

loss of valuable higher-order interactions that could otherwise be effectively captured by hypergraph neural networks. Based on the above discussion, our work makes the following key contributions:

- We propose a novel hypergraph representation for relational data that preserves both intra-tuple and inter-tuple relationships. Unlike traditional methods relying exclusively on primary key–foreign key (PK–FK) constraints, our representation decomposes tuples into attribute–value pairs, creating nodes naturally connected via hyperedges. This approach effectively captures fine-grained attribute-level interactions and is inherently schema-agnostic, eliminating the need for manual feature engineering or explicit schema knowledge. To the best of our knowledge, this is the first hypergraph-based representation specifically developed for relational database learning.

- We introduce rel-HNN, a hypergraph neural network specifically tailored for relational databases. Leveraging our hypergraph structure, rel-HNN learns explicit embeddings at three granularity levels—attribute–value pairs, tuples, and entire tables—thus effectively capturing both localized and global relational patterns and enabling richer relational learning.

- Additionally, to overcome the challenges posed by large-scale relational databases, we propose a split-parallel hypergraph learning algorithm that leverages multi-GPU parallelism. Our method enables full-hypergraph training by partitioning both data and computation across GPUs while preserving global structural context. Unlike mini-batch GNN training, which introduces redundant data movement and overlooks neighborhood completeness, our approach ensures efficient and context-aware learning. To the best of our knowledge, this is the first work to introduce split-parallelism for scalable training of hypergraph neural networks.

- Finally, our extensive experimental results demonstrate that the proposed multi-level representation framework enables rel-HNN to significantly outperform the state-of-the-art methods on both classification and regression tasks across a diverse set of real-world relational datasets. Moreover, our split-parallel training framework delivers substantial performance improvements on large-scale relational datasets and large benchmark hypergraph datasets, achieving speedups of up to $3.18\times$ and $2.94\times$, respectively, although the benefits of parallelization diminish for smaller datasets due to communication overhead. These results highlight the effectiveness and scalability of our approach.

The remainder of the paper is organized as follows. Section 2 introduces the preliminaries and formally defines the problem. Section 3 reviews relevant related work. Section 4 details the proposed methodology. Section 5 presents the experimental setup, results, and analysis. Finally, Section 6 concludes the paper and outlines potential directions for future work.

## 2 Background

A relational database (RDB) is defined as a collection of tables, denoted by $RDB = \{T^1, T^2, \ldots, T^n\}$, where each $T^k \in RDB$ represents a table. Each table captures information about a specific entity type, with rows (tuples) corresponding to entity instances and columns representing attributes or features. Table columns may contain diverse data types, including numerical values, categorical text, timestamps, geographic coordinates, and multimedia content. We denote the $i$-th row of table $T^k$ as $T_i^k$, and the $j$-th column as $T_{:j}^K$. Let $Attr^{T^k}$ denote the set of attributes (i.e., columns) of table $T^k$. An RDB is referred to as "relational" because values in a column $T_{:j}^k$ of one table may refer to rows in another table $T^m \in RDB$. Such columns are known as foreign keys and serve as the basis for modeling inter-table relationships. Learning tasks on a relational database ($RDB$) are typically formulated as predicting the values of a specific column in a designated target table. Let $T^{tg} \in RDB$ be the target table. In the training data, the rows of table $T^{tg}$ in the training set are associated with a label. The goal is to predict the labels for the remaining rows of table $T^{tg}$ where the label values are unknown. A hypergraph is a generalization of a traditional graph that can be represented as $H = (V, E, X)$, where $V = \{v_1, v_2, \ldots, v_{|V|}\}$ is the set of nodes or vertices, $E = \{e_1, e_2, \ldots, e_{|E|}\}$ denotes the set of hyperedges, and $X \in \mathbb{R}^{|V| \times d}$ is the feature matrix. Each hyperedge $e_i \subseteq V$ includes the set of nodes it connects.

## 3 Related Works

In this section, we review research closely related to our work. Section 3.1 discusses graph-based learning approaches for relational data, and Section 3.2 covers distributed multi-GPU training strategies for scalable graph and hypergraph learning.

### 3.1 Learning on *RDB* using Graphs

Graphs are well-suited for representing structured data, as they naturally capture relationships and dependencies among entities. Leveraging this property, graph neural networks (GNNs) have been applied to tabular data learning (Li et al., 2025). Here, the data instances (rows) are represented as nodes, and edges are formed based on similarities between instances using their feature vectors, capturing correlations between samples (Errica, 2023). In another line of work, each feature (column) is modeled as a node, and edges encode correlations or dependencies between features, which can be derived from statistical measures or learned feature embeddings (Zheng et al., 2023). The ability of GNNs to capture complex relational dependencies makes them a strong candidate for modeling relational data beyond tabular settings. Accordingly, GNNs have been increasingly adopted in relational database systems-related tasks (Li et al., 2025), including performance prediction (Zhou et al., 2020), query optimization (Song et al., 2022), and cardinality estimation (Chronis et al., 2024). Beyond these system-level tasks, GNNs have also been employed for supervised learning on relational databases (Cvitkovic, 2020a), where the database is represented as a graph to predict target attributes or classify tuples while preserving the underlying relational structure. A common approach for relational data in machine learning is to "flatten" relational data into a single table format, as most widely used supervised learning models require inputs in the form of fixed-size vectors. However, this flattening process often eliminates valuable relational information inherent in the data. The relational database is transformed into a graph, representing the relational structure, to predict the target attribute. In the graph representation, each node corresponds to a tuple, and edges represent foreign key relationships between these tuples. To manage the complexity of the graph, connections are typically restricted to a specific depth or number of hops. Finally, a GNN on this graph is applied to predict the target attribute using the GNN. A complete mapping between relational database concepts and graph terminology is provided in Appendix A. Representing a relational database (RDB) as a graph enables supervised learning tasks on RDBs to be framed as node classification problems (Atwood & Towsley, 2016). This representation supports both classification and regression tasks. The graph-based perspective also suggests that Graph Neural Network (GNN) techniques can be effectively applied to learning tasks involving relational databases. GNNs, particularly those designed for supervised learning on RDBs, commonly utilize the message passing framework (Gilmer et al., 2017). In this framework, each node $v$ in a graph is initially assigned a hidden representation $h_v^0$, which is iteratively updated over $R$ rounds of message passing. In each round $r$, node $v$ transmits a message $m_{vw}^r$ to each of its neighboring nodes $w$. These messages are generated by a learnable function that may incorporate edge features and depend on the current hidden states $h_v^r$ and $h_w^r$ of the source and target nodes, respectively. Each node then aggregates the messages received from its neighbors using another learnable function, resulting in an updated hidden state $h_v^{r+1}$. After $R$ rounds of message passing, a readout function aggregates the final hidden states $h_v^R$ across all nodes to produce a prediction for the entire graph. Learning with GNNs typically requires multiple message passing among nodes, which leads to significantly high training and inference times. To address this limitation, a recent approach named SPARE (Hilprecht et al., 2023) introduces an alternative encoding technique that represents the *RDB* as a directed acyclic graph (DAG). This DAG is constructed by performing a breadth-first search (BFS) on the undirected graph $G_t$ rooted at a target tuple $t \in T^{tg}$. Edges are directed from nodes at greater depth to nodes at lesser depth. This DAG structure enables single-pass learning, allowing for faster training and inference compared to standard GNNs. More recently, RelGT (Dwivedi et al., 2025) proposes a graph transformer architecture tailored for relational databases, capturing relational heterogeneity and long-range dependencies through multi-element tokenization and local–global attention. Both GNN- and DAG-based methods model relational data as standard graphs, often overlooking structural regularities defined by the database schema. This can lead to redundant computations and reduced learning efficiency. In contrast, a more recent approach, ATJ-Net, introduces a hypergraph-based representation that enables more adaptable and automated learning across relational databases (Bai et al., 2021).

Following the success of graph neural networks, hypergraph neural network models have received growing attention. HGNN (Feng et al., 2019) and HyperGCN (Yadati et al., 2019) extend graph convolutional networks to the hypergraph setting, while AllSet (Chien et al., 2022) introduces a two-stage message-passing framework. In this approach, hyperedge representations are first computed by aggregating the embeddings of their constituent nodes from the previous layer; then, the embeddings of the hyperedges connected to a node are aggregated to update that node's representation. Similar to graph-based approaches, research has been conducted on hypergraph-based tabular data learning also. HYTREL (Chen et al., 2023) is a hypergraph-enhanced tabular representation learning model in which each cell of a table is represented as a node, while rows, columns, and the entire table are modeled as distinct hyperedges. This design enables HYTREL to capture higher-order interactions among table components and preserve structural invariances. However, HYTREL primarily focuses on intra-table dependencies and does not model relational dependencies, making it unsuitable for complex relational databases. Furthermore, the substantial memory and computational overhead introduced by its attention-based hypergraph transformer, combined with the sparsity of the constructed hypergraph, limits its scalability to large tables. ATJ-Net(Bai et al., 2021) models the relational database $RDB$ as a hypergraph, each tuple $T_t^k$ of table $T^k \in RDB$ is considered as a hyperedge, that is $E = \{T_i^k\}_{k,i}$ where $T_i^k$ denotes the $i - th$ tuple of table $T^k \in RDB$. Joinable attributes serve as the vertices, and other attributes are treated as features of the hyperedges. Every table—except the main table—must include at least one joinable attribute; otherwise, it cannot be linked to the target label. If a hyperedge contains a vertex, a connection is established between them. Formally, let $Attr^{T^k}(j)$ be the value set of $j - th$ joinable attribute of table $T^k$ and $V = \{Attr^{T^k}(j)\}_{k,j}$. Moreover, the hypergraph can be viewed as a bipartite graph, with tuples and joinable attributes represented as two distinct sets of vertices, and edges indicating their inclusion relationships. ATJ-Net employs the message passing neural network (MPNN) framework (Gilmer et al., 2017) to construct a GNN over the heterogeneous hypergraph. In each GNN layer, hyperedge features are first aggregated to the vertices, which are then updated and propagated back to the hyperedges. Although ATJ-Net uses a hypergraph-based formulation, it ultimately trains a GNN by stacking standard GNN layers, which limits its capacity to fully exploit the rich relational structures across tables in $RDB$. Additionally, it considers only joinable attributes and neglects the representation learning of other attributes, potentially reducing its effectiveness in modeling complex intra-tuple associations.

## 3.2 Distributed multi-GPU GNN training

The challenge of scaling to large graphs has led to extensive research on multi-GPU GNN training systems. Training strategies for utilizing multiple GPUs have been devised for both mini-batch and full-graph training (Wang et al., 2022; Wan et al., 2023). In mini-batch multi-GPU training, nodes are first sampled into micro-batches, one for each GPU. Then, the local $k$-hop neighborhoods of the target nodes in each micro-batch are sampled and loaded. Each GPU subsequently trains the model on its assigned micro-batch in a data-parallel fashion (Wang et al., 2019; Gandhi & Iyer, 2021). However, data-parallel training often results in redundant data loading and computation due to overlapping $k$-hop neighborhoods across micro-batches. To mitigate this issue, GSplit (Polisetty et al., 2025) partitions the mini-batch into non-overlapping splits, assigning each split to a specific GPU. Each GPU processes only the vertices in its assigned split and exchanges intermediate results at each GNN layer. Nonetheless, mini-batch GNN training relies on neighborhood sampling, which may exclude important neighbors, leading to information loss and suboptimal message propagation. Full-graph multi-GPU training (Wang et al., 2022; Wan et al., 2023), in contrast, maintains the complete graph structure during training. G3 (Wan et al., 2023) splits the full graph into non-overlapping partitions, with each GPU responsible for one partition. After each layer, GPUs exchange outputs with other GPUs that require them for the next round of computation. While these methods are effective for large-scale graphs, they are not directly compatible with hypergraphs.

# 4 Proposed Methods

In this section, we present our proposed algorithms for learning from relational databases. We begin by constructing a hypergraph representation of the given relational database, which preserves fine-grained relationships and facilitates hypergraph neural network learning (Section 4.1). After transforming the relational database into a hypergraph, we apply our proposed model, rel-HNN, which captures complex relationships at

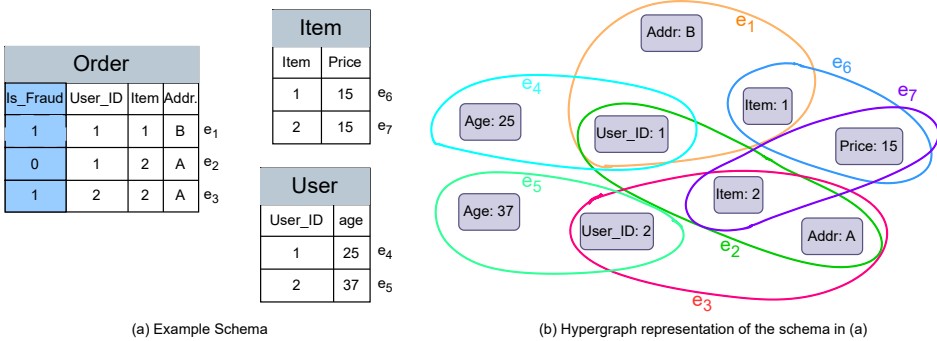

Figure 1: An example hypergraph generation for a relational database

multiple levels of granularity (Section 4.2). Finally, in Section 4.3, we introduce our split-parallel hypergraph learning algorithm designed to scale effectively to large relational databases.

## 4.1   Relational Database to Hypergraph

We now detail our hypergraph construction approach for relational databases, which enables hypergraph neural network learning by capturing complex, fine-grained intra-tuple relationships beyond conventional primary key–foreign key constraints. Our approach decomposes each tuple in the RDB into attribute-value pairs. Instead of representing a tuple or row as a node as existing GNN-based approaches, we create a node for each unique attribute-value pair, $(\text{Attr}_j^k, T_{i,j}^k)$, found in all the tables contained by the database. Then, for each table $T^k \in \text{RDB}$, for each row $T_{i,:}^k$, we create a hyperedge that connects the nodes associated with the attribute-value pairs, $(\text{Attr}_j^k, T_{i,j}^k)$, contained by the row. This hypergraph formulation provides a natural way to model fine-grained interactions within tuples, moving beyond rigid primary key–foreign key constraints, enabling the learning of richer representations. Figure 1 illustrates a hypergraph representation for a relational database, where attribute–value pairs are represented as nodes. Edges are shown as colored circles, each connecting more than two nodes to form a hyperedge. For the feature vector $X_v$ of a node $v \in V$, we consider two techniques. In the first approach, we assign one-hot encoded vectors as feature representations for the nodes. In the second approach, we construct the feature vector such that each index corresponds to either an attribute from the tables or a value present in the tables. For each node, we initialize a zero vector and set the indices corresponding to its associated attribute and value to 1. The detailed construction procedure is provided in Algorithm 1 in Appendix C.

## 4.2   rel-HNN: Hypergraph Neural Network for Relational data

Building on the hypergraph construction described in the previous subsection, we now present rel-HNN, our proposed hypergraph learning algorithm designed to operate on the resulting hypergraph derived from a relational database. Rel-HNN adopts a two-phase message-passing mechanism (Chien et al., 2022) that alternates between aggregating information from nodes to hyperedges and then from hyperedges back to nodes. In each layer, for hyperedge embedding, messages (i.e., embeddings) are aggregated from the nodes it connects. Similarly, each node updates its representation by receiving messages from the hyperedges to which it belongs. These two phases naturally capture the semantics of relational joins: the first phase aggregates attribute-level information within each relation, while the second phase propagates the relational context back to the corresponding attribute-value pairs. In addition to the prior two-phase hypergraph learning techniques, rel-HNN introduces a relation-aware, table-conditioned aggregation mechanism. We integrate learnable table embeddings and relation-specific transformations, which enable rel-HNN to jointly capture structural dependencies and relational semantics across heterogeneous hyperedges within relational databases. Figure 2 illustrates the architecture of rel-HNN as applied to the hypergraph depicted in Figure 1.

At the initial layer (layer 0), we compute the embedding $Z_v^0$ of a node $v \in V$ by applying a multilayer perceptron $MLP_V^0$ to its input feature vector $X_v$, that is $Z_v^0 = MLP_V^0(X_v)$. Learning node embeddings that

represent attribute–value pairs allows the model to capture fine-grained semantic relationships within tuples, resulting in more expressive representations of relational data. Subsequently, for each hyperedge $e \in E$, we determine its intermediate embedding at the initial layer, denoted as $F_e^0$, by applying another multilayer perceptron to the sum of the initial node embeddings it connects, as defined in Equation 1.

$$F_e^0 = MLP_E^0 \left( \sum_{v \in e} Z_v^0 \right) \tag{1}$$

In Figure 2, the initial embedding $F^0 e_4$ of edge $e_4$ is determined by applying the multilayer perceptron on the sum of the initial embeddings of node $V_{Age:25}$ and node $V_{User_ID:1}$, i.e., $F^0 e_4 = MLP^0 E(Z_{V_{Age:25}} + Z_{V_{User_ID:1}})$, as edge $e_4$ is connected to these two nodes. By aggregating node embeddings to learn hyperedge embeddings that represent tuples, the model encodes higher-order interactions and co-occurrence patterns among attribute–value pairs, providing a comprehensive understanding of tuple-level semantics.

For each hyperedge $e \in E$, we learn its final embedding at $layer : 0$ by concatenating its intermediate embedding $F_e^0$ with $Z_{T_e}$, the embedding of the table that contains the row corresponding to the hyperedge, which is $Z_e^0 = \text{CONCAT}(F_e^0, Z_{T_e})$. In Figure 2, the final embedding of edge $e_4$ at layer 0, $Z^0 e_4$, is obtained by concatenating the initial embedding of $e_4$, $F^0 e_4$, with the table embedding of table $T^{User}$, as $e_4$ originates from the $User$ table; i.e., $Z^0 e_4 = F^0 e_4 || ZT^{User}$. In our model, learning table-level embeddings enables the incorporation of global patterns shared across all tuples within a table.

For the intermediate layers (layer 1 to layer L), we determine the embedding of a node $v$, $Z_v^l$, by aggregating the hyperedge embeddings from the previous layer (Equation 2). Here, $MLP_v^l$ is the multi-layer perceptron for nodes at layer $l$ and $\mathcal{E}_v$ is the set of hyperedges in $E$ that contains $v$.

$$Z_v^l = MLP_V^l \left( \sum_{e \in \mathcal{E}_v} Z_e^{l-1} \right) \tag{2}$$

For node $V_{User\_id:1}$ in Figure 2, the embedding at layer 1 is determined by applying the multilayer perceptron $MLP_V^0$ to the summation of the final embeddings at layer 0 of hyperedges $e_4$, $e_1$, and $e_2$, as these hyperedges include node $V_{User\_id:1}$. Similarly, we determine the embedding of a hyperedge $e$ at layer $l$, denoted as $Z_e^l$, by aggregating the node embeddings from the previous layer, as defined in Equation 3. Here, $MLP_E^l$ refers to the multilayer perceptron applied to hyperedges at layer $l$. Note that, through shared attribute–value pairs, the model is able to capture complex relationships between tuples both within and across tables.

$$Z_e^l = MLP_E^l \left( \sum_{v \in e} Z_v^l \right) \tag{3}$$

For each hyperedge $e$ corresponding to a row in the target table $T^{tg}$, the final embedding $Z_e^L$, where $L$ is the last layer, represents the predicted class probability. In Figure 2, the hyperedges $e_1$, $e_2$, and $e_3$ correspond to rows of the target table $T^{Order}$. The pseudocode of the rel-HNN algorithm is provided in Algorithm 2 in Appendix C.

### 4.3 Split-Parallel Hypergraph Learning for large databases

To enable scalable learning on large hypergraphs derived from real-world relational databases, we propose a split-parallel hypergraph neural network algorithm. In practice, relational databases tend to be large, and converting them into hypergraphs results in a substantial number of nodes and hyperedges. In particular, assigning a node to each attribute–value pair significantly increases the number of nodes, especially in sparse datasets. Learning on such large hypergraphs introduces critical challenges in terms of computational runtime and memory usage, often making it infeasible to load the entire hypergraph into the limited memory of a single GPU. To address these limitations, we partition the hypergraph by dividing the node set into disjoint splits and distribute the computation of each split and its incident hyperedges across multiple GPUs. This

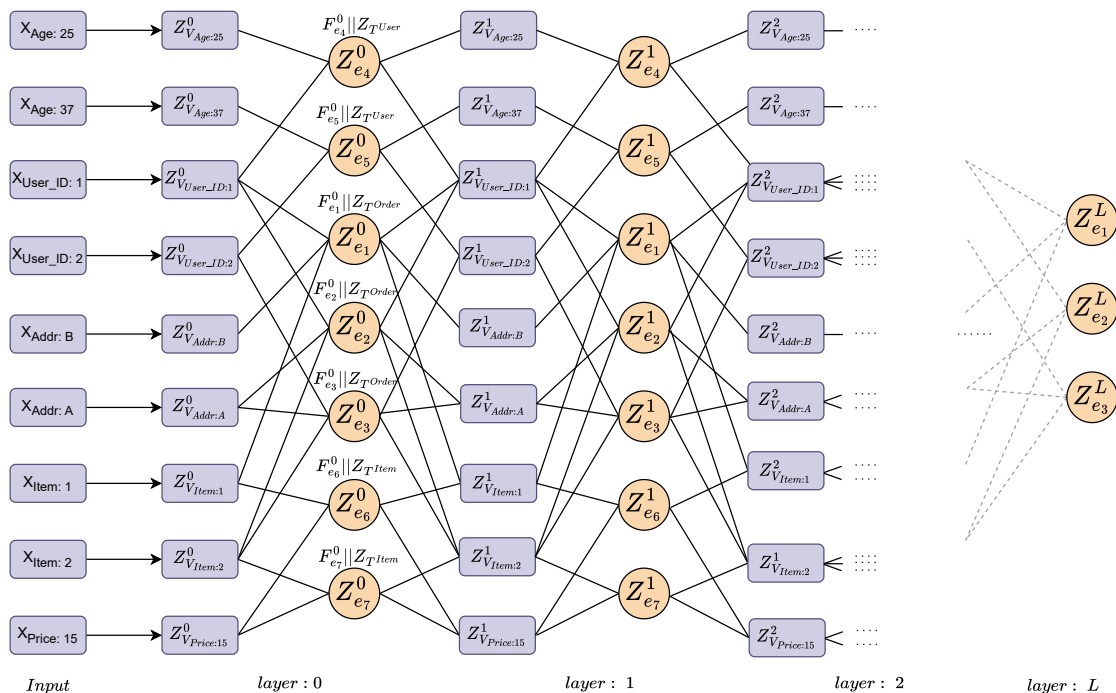

Figure 2: HNN architecture for the hypergraph shown in Figure 1

strategy effectively mitigates both memory and runtime bottlenecks, making the training process practical for large-scale relational databases. However, learning from node-partitioned hypergraphs is non-trivial, as hyperedge embeddings rely on aggregating node embeddings that may be distributed across multiple GPUs, leading to communication overheads and synchronization challenges during training.

To utilize parallel processing, assuming there are $N$ GPUs available, we divide the nodes $V$ in the hypergraph $H = (V, E, X)$ into $N$ partitions, $V_1, V_2, \ldots$, and $V_N$. Following the partition, we split the hypergraph $H = (V, E, X)$ into $N$ partitions, $H_1 = (V_1, E_1, X_1)$, $H_2 = (V_2, E_2, X_2)$, $\ldots$, and $H_N = (V_N, E_N, X_N)$. Here, $E_i = \{\{v | v \in e \text{ and } v \in V_i\} | e \in E \text{ and } e \cap V_i \neq \emptyset\}$. Given $N$ GPUs ($GPU_1, GPU_2,, \ldots,$ and $GPU_N$), we load the hypergraph $H_i$ into $GPU_i$. Now, in each GPU, $GPU_i$, in parallel, we determine the embedding of a node $v \in V_i$ at layer 0, $Z_v^0$, by applying the same multilayer perceptron layer $MLP_V^0$ to the feature vector $X_v$, similar to rel-HNN. The pseudocode of our proposed split parallel Hypergraph Neural Network algorithm is provided in Algorithm 3 in Appendix C. In Figure 3, we demonstrate an example of parallel learning using two GPUs (GPU-1 and GPU-2). We divide the nodes in two partition, node $V_1 = \{V_{Age:25}, V_{Age:37}, V_{User\_ID:1}, V_{User|ID:2}, V_{Addr:B}\}$ and $V_2 = \{V_{Addr:A}, V_{Item:1}, V_{Item:2}, V_{Price:15}\}$. The local computations are visualized in the blue-shaded area for $GPU_1$ and the red-shaded area for $GPU_2$. For each node, the embedding at layer 0 is learned locally on the assigned GPU. For example, the embedding for $V_{Age:25}$ is learned in GPU-1.

In rel-HNN, we apply the multilayer perceptron layer, $MLP_E^0$, to the sum of node embeddings to determine the intermediate embedding at the initial layer, $F_e^0$ (Equation 1). For split learning, we divide $MLP_E^0$ into two components, consisting of a linear transformation $MLP_{E_\mathcal{L}}^0$ followed by a non-linear transformation function $MLP_{E_\sigma}^0$. Let the original $MLP_E^0$ consist of L stacked layers defined recursively as $H_1(x) = \sigma_1(W_1 x + b_1), H_2(x) = \sigma_2(W_2 H_1(x) + b_2), \ldots, H_L(x) = \sigma_L(W_L H_{L-1}(x) + b_L)$. Then, $MLP_E^0(x) = H_L(x)$. For split-parallel learning, the first linear mapping can be applied independently to the local embeddings on each GPU, which is $MLP_{E_\mathcal{L}}^0(x) = W_1 x$. The subsequent transformations and activation functions are grouped into $MLP_{E_\sigma}^0(x) = \sigma_L(W_L \sigma_{L-1}(W_{L-1} \ldots \sigma_2(W_2 (\sigma_1(x + b_1)) + b_2) \cdots + b_{L-1}) + b_L)$, which is applied after the local linear projections are aggregated across GPUs. A justification of this decomposition is provided

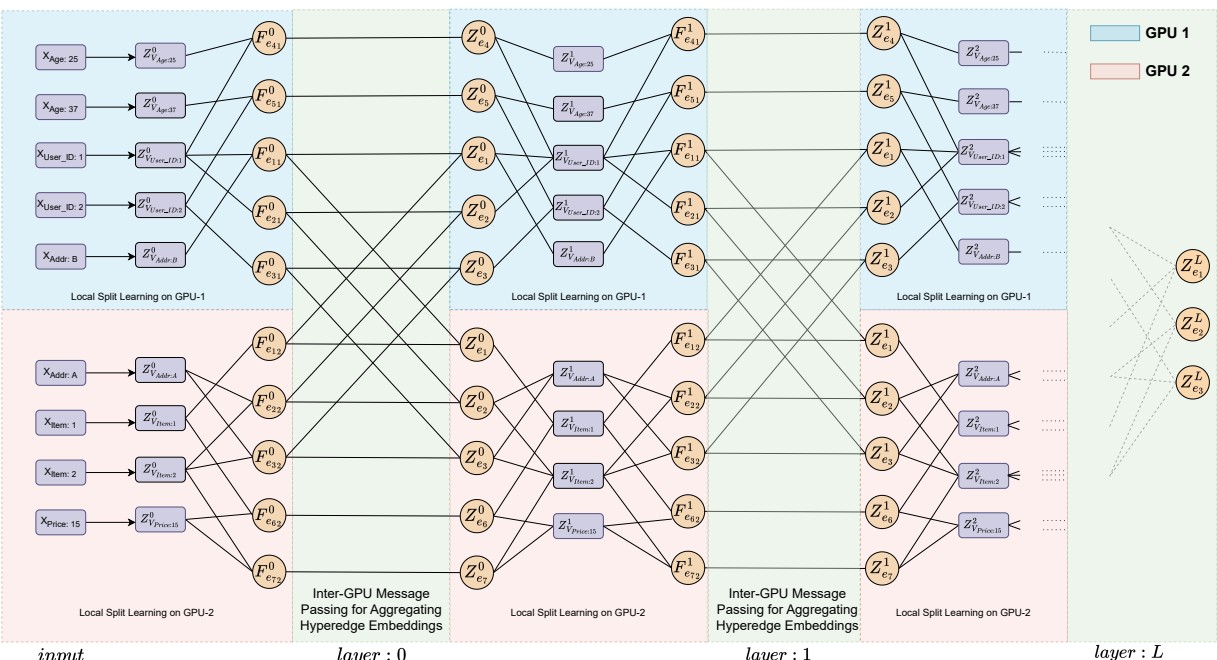

Figure 3: Split-Parallel learning of Hypergraph Neural Network (HNN) on two different GPUs. Blue and red shaded area represent the local processes of GPU-1 and GPU-2 respectively.

in Appendix D. We compute the local hyperedge embeddings at the initial layer associated with the $i$-th partition, denoted as $F^0_{e_i}$, by applying $MLP^0_{E_{\mathcal{L}}}$ to the sum of the node embeddings at layer 0 (Equation 4).

$$F^0_{e_i} = MLP^0_{E_{\mathcal{L}}} \left( \sum_{v \in e \cap V_i} Z^0_v \right) \tag{4}$$

In Figure 3, the local hyperedge embedding at layer 0 for hyperedge $e_1$ at GPU-1, $F^0_{e_{1_1}}$, is learned from $V_{User\_ID:1}$ and $V_{Addr:B}$, the constituent nodes in the partition. On the other hand, for GPU-2, $F^0_{e_{1_2}}$ is learned from $V_{Item:1}$.

To learn the global hyperedge embeddings at the initial layer, $F^0_e$, we determine the sum of the local embeddings and apply the non-linear activation function $MLP^0_{E_\sigma}$ (Equation 5). This aggregation of local embeddings from different GPUs requires inter-GPU message passing. We allocate a global tensor with $|E|$ rows with the same global ordering of edges on every GPU. Each GPU fills its entries for edges in its partition with partial vectors (zeros elsewhere) and then performs an AllReduce (sum) operation across GPUs to obtain the global embeddings.

$$F^0_e = MLP^0_{E_\sigma} \left( \sum_{i=0}^N F^0_{e_i} \right) \tag{5}$$

Finally, for each hyperedge $e \in E$, we learn its final embedding at layer 0, $Z^0_e$, by concatenating $F^0_e$ with $Z_{T_e}$, the embedding of the corresponding table. In Figure 3, the final embedding of $e_1$ at layer 0, $Z^0_{e_1}$, is learned by aggregating their local embeddings at GPU-1 and GPU-2 and then concatenating their corresponding table embedding, $Z_{T_{Order}}$.

For the intermediate layers, we determine the embedding of a node $v \in V_i$, $Z^l_v$, using Equation 2, in parallel. The process of determining node embeddings requires no inter-GPU messaging as the corresponding hyperedge embeddings are already accumulated in the assigned GPU. For example, in Figure 3, the embed-

ding of node $V_{User\_ID:1}$ at layer 1, $Z^1_{V_{User\_ID:1}}$, is calculated by aggregating $Z^0_{e_1}$ and $Z^0_{e_2}$ which are already accumulated in GPU-1. For the hyperedge embeddings, similar to the initial layer, we divide $MLP^l_E$ as a single hidden layer MLP consisting of one linear transformation, $MLP^l_{E_{\mathcal{L}}}$, followed by remaining non-linear transformations $MLP^l_\sigma$. We determine the local hyperedge embeddings at the initial layer associated with the $i$-th partition, $F^l_{e_i}$, by applying $MLP^l_{E_{\mathcal{L}}}$ on the sum of the embeddings of the nodes from the same layer (Equation 6).

$$F^l_{e_i} = MLP^l_{E_{\mathcal{L}}} \left( \sum_{v \in e} Z^l_v \right) \tag{6}$$

Finally, we learn the global hyperedge embeddings at layer $l$, $Z^l_e$, we determine the sum of the local embeddings and apply the non-linear activation function $MLP^l_{E_\sigma}$ (Equation 7).

$$Z^l_e = MLP^l_{E_\sigma} \left( \sum_{i=0}^{N} F^l_{e_i} \right) \tag{7}$$

## 5 Empirical Evaluation

In this section, we present a comprehensive set of experiments focused on finding the effectiveness and performance gain of our proposed rel-HNN approach. The structure of this section is as follows: we begin by detailing the datasets used and addressing model architecture, along with the experimental settings. Next, we separately demonstrate the performance improvements achieved by our approach on both classification and regression tasks. Lastly, we analyze the effectiveness of our split parallel hypergraph learning approach on both relational and hypergraph datasets.

### 5.1 Experiment Design

To verify the effectiveness of our proposed rel-HNN model, we conducted experiments on both classification and regression tasks. We compared its performance against state-of-the-art graph-based algorithms for learning on relational data. Specifically, we applied GCN (Kipf & Welling, 2017) and GAT (Velickovic et al., 2018) on graphs constructed from tuples connected through primary key–foreign key (PK–FK) relationships. For SPARE (Hilprecht et al., 2023), we considered both GCN- and GAT-based variants, referred to as SPARE-GCN and SPARE-GAT, respectively. For hypergraph neural networks, we have included HyperGCN (Feng et al., 2019), HGNN (Yadati et al., 2019), and DPHGNN (Saxena et al., 2024). We also evaluated ATJ-Net (Bai et al., 2021), which leverages a heterogeneous GNN architecture. We experimented with RelGT (Dwivedi et al., 2025), a graph transformer architecture developed specifically for relational databases . In our experiments, we included four versions of rel-HNN. Rel-HNN-one uses one-hot encoding, while rel-HNN-av uses $attribute-value$ encoding for node features. In both rel-HNN-one and rel-HNN-av, table embeddings are omitted. In contrast, rel-HNN-one-t and rel-HNN-av-t include learnable table embeddings in their respective architectures. For all the rel-HNN variants, we have set the number of layers $L = 2$. The embedding length of all the nodes and hyperedges is fixed at two. For rel-HNN-one-t and rel-HNN-av-t, the table embedding dimension is set to 8 across all the datasets. We have adopted stratified 5-fold cross-validation to preserve class distribution. All experiments were conducted on a workstation equipped with an Intel Core i7-7700 CPU @ 3.60GHz, 48GB RAM, and four NVIDIA GeForce RTX 3060 Ti GPUs with 8GB of memory each. To validate the applicability in real relational database environments, we developed an end-to-end pipeline connecting a PostgreSQL database to the rel-HNN training framework also. The pipeline extracts tuples and relationships using lightweight SQL queries, transforms them into hypergraph representations without requiring costly database operations as joins.

### 5.2 Performance on Classification Tasks

Table 1 presents the AUROC scores for different methods evaluated across the nine real-life datasets (see Appendix E for detailed dataset descriptions). The results clearly show that the rel-HNN variants consistently outperform the baseline methods with the highest AUROC score on nine out of nine datasets while

maintaining comparable or lower standard deviations, indicating stable performance. Among the state-of-the-art methods, RelGT performs the best on six out of the nine datasets. On the remaining three datasets, ATJ-net performs the best. However, on datasets such as *SameGen*, and *rel-f1(dnf)*, ATJ-net exhibits comparatively lower performance than the GCN, GAT, and SPARE variants. Among the GNN-based approaches, SPARE variants generally achieved a slightly higher AUROC score compared to GCN and GAT. These graph-based methods model tuples as monolithic nodes and primarily rely on primary key–foreign key relationships, which limits their ability to capture fine-grained relational dependencies. Within the class of hypergraph neural network-based methods, DPHGNN outperforms both HyperGCN and HGNN. However, these models are designed for generic hypergraphs and treat hyperedges as homogeneous sets, lacking explicit relational or table-level semantics. In addition, HyperGCN relies on graph-based approximations of hyperedges, which limits the modeling of higher-order interactions, while HGNN and DPHGNN employ fixed aggregation schemes that ignore varying hyperedge roles.

Table 1: Performance comparison of methods across datasets for classification tasks. Mean AUROC is shown with standard deviation in parentheses; bold values indicate the best performance.

| Method | Hepa | Pima | Cora | SameGen | IMDB | Mutag | rel-f1 (top3) | rel-f1 (dnf) | rel-avito (uv) |
|---|---|---|---|---|---|---|---|---|---|
| GCN | 0.5484 (0.0820) | 0.5438 (0.0746) | 0.5365 (0.0502) | 0.5245 (0.0918) | 0.6621 (0.0854) | 0.6357 (0.0650) | 0.5582 (0.0192) | 0.5122 (0.0837) | 0.4751 (0.0531) |
| GAT | 0.5451 (0.0465) | 0.5442 (0.0746) | 0.5393 (0.0529) | 0.5119 (0.0509) | 0.6632 (0.0364) | 0.6320 (0.0683) | 0.5589 (0.0283) | 0.5064 (0.0928) | 0.4801 (0.0850) |
| SPARE-GCN | 0.5636 (0.0374) | 0.5511 (0.0374) | 0.5712 (0.0650) | 0.5233 (0.0465) | 0.6718 (0.0827) | 0.6044 (0.0912) | 0.5657 (0.0928) | 0.5241 (0.0918) | 0.5085 (0.0537) |
| SPARE-GAT | 0.5594 (0.0465) | 0.5527 (0.0509) | 0.5679 (0.0370) | 0.5276 (0.0746) | 0.6865 (0.0991) | 0.6271 (0.0928) | 0.5323 (0.0501) | 0.5222 (0.0703) | 0.5101 (0.0608) |
| HyperGCN | 0.5075 (0.0426) | 0.4983 (0.0394) | 0.4833 (0.0364) | 0.5134 (0.0401) | 0.7348 (0.0273) | 0.5832 (0.0323) | 0.5213 (0.0273) | 0.4923 (0.0099) | 0.5056 (0.0679) |
| HGNN | 0.4997 (0.0375) | 0.5032 (0.0021) | 0.4819 (0.0178) | 0.5087 (0.0628) | 0.7483 (0.0195) | 0.5783 (0.0279) | 0.5012 (0.0933) | 0.4753 (0.1028) | 0.5148 (0.0828) |
| DPHGNN | 0.5482 (0.0568) | 0.5238 (0.0546) | 0.5247 (0.0808) | 0.5439 (0.0528) | 0.7526 (0.0183) | 0.5915 (0.0839) | 0.5236 (0.0274) | 0.5023 (0.0368) | 0.5313 (0.0916) |
| ATJ-net | 0.5950 (0.0444) | 0.6072 (0.0314) | 0.6242 (0.0111) | 0.5030 (0.0078) | 0.8575 (0.0273) | 0.8812 (0.0505) | 0.6412 (0.0081) | 0.5030 (0.0078) | 0.5625 (0.0942) |
| RelGT | 0.6643 (0.0340) | 0.5839 (0.0384) | 0.6513 (0.0641) | 0.6464 (0.0339) | 0.8450 (0.0189) | 0.7215 (0.0126) | 0.8234 (0.0682) | 0.7542 (0.0239) | 0.6874 (0.0836) |
| rel-HNN-one | 0.8264 (0.0776) | **0.7158** (0.0180) | **0.7364** (0.0582) | 0.8096 (0.0642) | 0.8791 (0.0103) | 0.8777 (0.0496) | 0.8602 (0.0193) | 0.7614 (0.0182) | 0.7413 (0.0131) |
| rel-HNN-av | 0.6515 (0.1485) | 0.7097 (0.0205) | 0.7041 (0.0336) | 0.8166 (0.0641) | 0.8728 (0.0160) | **0.8976** (0.0560) | 0.8553 (0.0118) | 0.7563 (0.0143) | 0.7363 (0.0283) |
| rel-HNN-one-t | **0.8916** (0.0596) | 0.7023 (0.0111) | 0.6521 (0.0825) | **0.8250** (0.0646) | **0.8814** (0.0105) | 0.8697 (0.0537) | **0.8685** (0.0263) | **0.7616** (0.0232) | **0.7712** (0.0117) |
| rel-HNN-av-t | 0.8667 (0.0287) | 0.6982 (0.0260) | 0.5119 (0.0239) | 0.8218 (0.0647) | 0.8732 (0.0238) | 0.7300 (0.2495) | 0.7541 (0.1261) | 0.7206 (0.0488) | 0.7418 (0.0718) |

Among the rel-HNN variants, rel-HNN-one-t (rel-HNN with one-hot encoding and table embedding) achieves the highest AUROC scores on most datasets, including *Hepa* (0.8916), *SameGen* (0.8250), *IMDB* (0.8814), *rel-f1 (top3)* (0.8685), *rel-f1 (dnf)* (0.7616), and *rel-avito (uv)* (0.7712). Rel-HNN-one-t demonstrates competitive AUROC performance on the remaining datasets as well. The standard deviations for this variant are also relatively low or comparable across datasets, indicating stable and consistent performance. Among the other three versions of rel-HNN, rel-HNN-av-t is a strong contender that outperforms the other two variants on datasets such as *Hepa*, *SameGen*, *IMDB*, and *rel-avito(uv)*. The benefit of learning table embeddings explicitly is evident from the performance gains of the *-t* variants. Both *rel-HNN-one* and *rel-HNN-av* exhibit notable improvements after the concatenation of the table embedding that represents the global information of all tuples in a table. An important observation is that in the datasets where the *non-t* variants perform relatively better such as, the number of tables is relatively small, suggesting that explicit table-level embeddings may be less beneficial when the relational structure is simple or shallow. For example, there

are only three tables in both *Cora* and *Mutag*, where the *non-t* variants perform better. For the dataset *Pima*, the number of tables is relatively higher (nine), but the performance gap in favor of non-t variants is also insignificant compared to *Cora* and *Mutag*. However, compared to existing state-of-the-art methods, rel-HNN-one and rel-HNN-av demonstrate substantial performance gains, despite not utilizing table-level embeddings.

The performance results presented in Table 1, when analyzed alongside the dataset statistics in Table 5, reveal several key trends. Across a diverse range of datasets, spanning small-scale ones such as *Pima* and *SameGen*, medium-scale datasets like *Hepa*, *Cora*, and *Mutag*, and large-scale datasets including *rel-f1 (top3)*, *rel-f1 (dnf)*, and *rel-avito (uv)*, our proposed methods consistently achieve superior performance. The relative performance gain of our models is substantially higher on smaller datasets such as *Hepa* (34.2%), *Pima* (17.9%), *Cora* (13.1%), and *SameGen* (27.6%), with *Mutag* (1.9%) being a notable exception. In contrast, the improvement is more moderate for larger datasets, including *IMDB* (2.78%), *rel-f1 (top3)* (5.5%), and *rel-f1 (dnf)* (1.0%), while *rel-avito (uv)* (12.2%), the largest dataset, deviates from this trend. This highlights a key limitation of existing state-of-the-art methods in coping with both extremes of data availability, struggling to generalize in data-scarce (small-scale) settings and to scale efficiently in large-scale environments. In summary, our methods adapt effectively to varying levels of schema complexity and data volume, making them suitable for both compact and large-scale relational databases. Rel-HNN-one-t emerges as the most robust model, delivering both high AUROC scores and low variance. These results collectively demonstrate the effectiveness of our proposed framework on classification problem across diverse relational datasets.

### 5.3 Performance on Regression Tasks

For regression tasks, we have collected five relational datasets (see Appendix E for detailed dataset descriptions). Table 2 reports the Root Mean Square Error (RMSE) performance across four regression datasets for various methods. Across all datasets, the proposed rel-HNN variants significantly outperform the state-of-the-art methods in terms of RMSE values. On the *Pyrimidine* dataset, which is relatively small, rel-HNN-one achieves the lowest RMSE of 0.0792, outperforming SPARE-GCN, the best-performing baseline method, by a substantial margin. Other variants of rel-HNN have also performed better than the existing approaches. In the *ClassicModels* dataset, which contains more tables and a larger schema, the RelGT performs the best among existing methods with an RMSE value of 537.7372. In contrast, the rel-HNN variants reduce the error significantly, with rel-HNN-av-t achieving an RMSE value of 115.8039. This demonstrates the better generalization capability of our models in complex relational structures. The improvements are even more pronounced in the *Pubs* dataset. Here, RelGT yields an RMSE of 72.4847, while all rel-HNN versions reduce the error drastically to around 6, with rel-HNN-one-t achieving the lowest RMSE of 5.4055. This suggests that the rel-HNN architecture is particularly effective in datasets with rich attribute columns and intricate relational dependencies. For the *Biodegradability* dataset, RelGT again achieves the best performance among prior approaches, with an RMSE of 18.4748. In contrast, *rel-HNN-one-t* significantly outperforms all baselines, achieving the lowest RMSE of 1.4779. Other *rel-HNN* variants also attain RMSE values around 1.5, demonstrating the scalability and effectiveness of our method in handling large and complex relational datasets. For the rel-f1 (position) dataset, RelGT substantially reduces the RMSE to 6.7498 compared to prior baselines, while rel-HNN-one-t attains the lowest RMSE of 3.4585.

Overall, the rel-HNN models consistently achieve lower RMSE values and greater stability across datasets, as indicated by smaller standard deviations in most cases. The different variants of rel-HNN demonstrate unique strengths. Rel-HNN-one achieves the lowest RMSE score on *Pyrimidine*, which consists of two tables only, suggesting its effectiveness for limited schema complexity. On the other hand, for larger and complex datasets with a higher number of tables, the table embedding variants (one-t and av-t) tend to perform better. Rel-HNN-one-t provides the best performance on *Pubs* (5.4055), *rel-f1 (position* (3.4585), and *Biodegradability* (1.4779), and near-best performance on *ClassicModels* (117.5294). Rel-HNN-av-t achieves the best performance on *ClassicModels* (115.8039) and is narrowly outperformed on *Pubs* (5.6332) by *rel-HNN-one-t*. For the *ClassicModels* dataset, although rel-HNN-av-t is outperformed by both rel-HNN-one and rel-HNN-one-t, it performs better than its non-table counterpart, rel-HNN-av. Overall, the flexibility among variants allows the rel-HNN framework to adapt effectively across a broad spectrum of relational learning tasks.

Table 2: RMSE Comparison Across Datasets in Regression Tasks

| Method | Pyrimidine | ClassicModels | Pubs | Biodegradability | rel-f1 (position) |
|---|---|---|---|---|---|
| GCN | 0.1195 ± 0.0591 | 955.4509 ± 90.8273 | 225.6450 ± 20.0912 | 18.8374 ± 5.3645 | 15.1948 ± 2.7599 |
| GAT | 0.1183 ± 0.0530 | 989.2736 ± 80.5091 | 231.1827 ± 16.7465 | 19.5109 ± 5.1827 | 15.6837 ± 2.4883 |
| SPARE-GCN | 0.1061 ± 0.0517 | 809.0918 ± 67.7364 | 125.6509 ± 12.9283 | 14.3746 ± 7.6509 | 15.3938 ± 2.3948 |
| SPARE-GAT | 0.1149 ± 0.0469 | 808.6547 ± 59.9182 | 142.8374 ± 15.6501 | 18.0928 ± 4.9821 | 15.3443 ± 3.1913 |
| HyperGCN | 0.1283 ± 0.1329 | 921.5709 ± 102.4283 | 130.8904 ± 18.3423 | 18.4759 ± 7.1238 | 15.4839 ± 3.0293 |
| HGNN | 0.1212 ± 0.1385 | 914.5710 ± 98.8957 | 108.4856 ± 17.6765 | 18.1492 ± 6.7865 | 15.3847 ± 2.9749 |
| DPHGNN | 0.1035 ± 0.0469 | 821.7203 ± 73.3498 | 107.2947 ± 12.9347 | 19.8453 ± 6.8463 | 14.3847 ± 2.4759 |
| ATJ-net | 0.1235 ± 0.0331 | 728.8391 ± 52.2696 | 95.3952 ± 7.1283 | 24.7605 ± 1.1405 | 11.2039 ± 2.3948 |
| RelGT | 0.1098 ± 0.2348 | 537.7372 ± 21.4374 | 72.4847 ± 8.2344 | 18.4748 ± 9.6134 | 6.7498 ± 1.9283 |
| rel-HNN-one | **0.0792 ± 0.0394** | 120.7200 ± 18.9351 | 6.0572 ± 1.0807 | 1.5909 ± 0.2871 | 3.7384 ± 0.3095 |
| rel-HNN-av | 0.0843 ± 0.0308 | 117.6724 ± 17.4639 | 6.0106 ± 1.1637 | 1.6729 ± 0.0198 | 3.8938 ± 0.3293 |
| rel-HNN-one-t | 0.0975 ± 0.0365 | 117.5294 ± 9.2348 | **5.4055 ± 1.4554** | **1.4779 ± 0.4093** | **3.4585 ± 0.2003** |
| rel-HNN-av-t | 0.0979 ± 0.0574 | **115.8039 ± 14.1657** | 5.6332 ± 1.5954 | 1.6390 ± 0.3934 | 3.6738 ± 0.2039 |

## 5.4 Split-Parallel Hypergraph Learning Performance

To observe the effectiveness of parallel processing, we conducted experiments using the split learning process on a classification task. All experiments were performed using the rel-HNN-one-t version of the rel-HNN model. We evaluated performance on two different sets of datasets. The first set consists of the same datasets used in Section 5.2. The second set includes widely used benchmark hypergraph datasets (see Appendix E for detailed dataset descriptions). Figure 4 presents the training time per epoch (in milliseconds) across multiple datasets under varying numbers of GPUs ($N$), reflecting the impact of split-learning and parallelism.

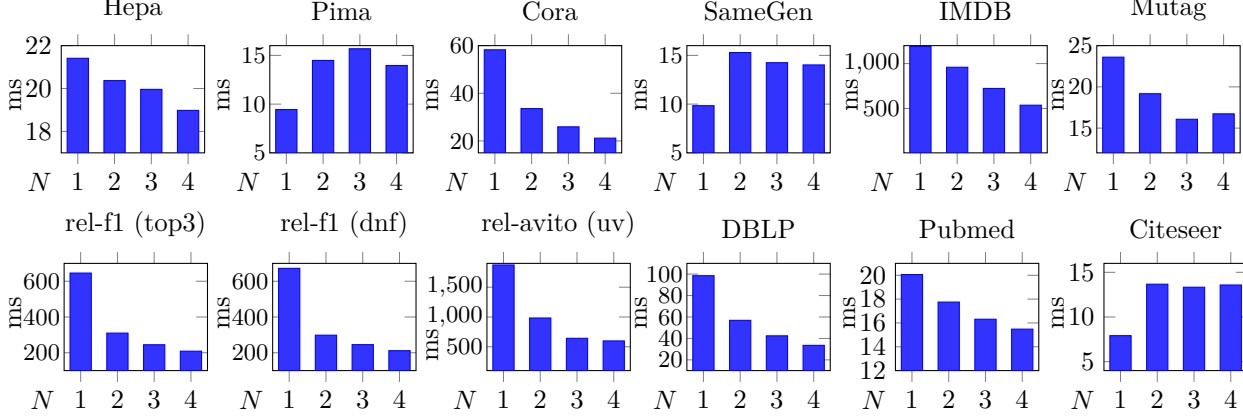

Figure 4: Training time per epoch (in milliseconds) across databases for different numbers of GPUs ($N$)

A clear trend is observed for larger datasets in terms of number of nodes as *Hepa*, *Cora*, *IMDB*, *Mutag*, *rel-f1 (top3)*, *rel-f1 (dnf)*, and *rel-avito (uv)*. As we increase the number of GPUs, the training time per epoch decreases substantially. For example, on the *rel-f1(dnf)* dataset, the training time per epoch drops significantly from 672.33 ms with a single GPU to 211.42 ms with four GPUs—a speedup of nearly 3.18x. This indicates the substantial benefit of parallel training for large datasets. The speedup tends to diminish as the dataset size decreases, primarily due to insufficient workload to fully utilize multiple GPUs, leading to communication overheads diminishing the benefits of parallelization. For example, on dataset *Mutag*, which is relatively smaller than *rel-f1(dnf)*, the speedup drops to 1.41x, with the training time per epoch decreasing from 23.60 ms to 16.73 ms. On much smaller datasets, such as *Pima* and *SameGen*, we observe that increasing the number of GPUs increases the training time per epoch, as we employ two GPUs. As the number of GPUs increases, the training time per epoch decreases, but fails to improve beyond the single-GPU performance. Here, the inter-GPU communication and processing overheads for aggregating hyperedge embeddings are outweighing the benefits of parallelization. Among hypergraph datasets, the most significant runtime reduction is observed on the largest dataset, *DBLP*, where training time drops from 98.51 ms with

a single GPU to 33.53 ms with four GPUs, achieving a speedup of nearly $2.94\times$. A similar decreasing trend is seen in *Pubmed*, which shows a consistent improvement as $N$ increases, reducing training time from 20.06 ms to 15.48 ms with a speedup of $1.30\times$. However, in the case of *Citeseer* with relatively fewer nodes, an increase in $N$ results in a rise in training time, likely due to parallelization overhead surpassing the computational benefits for smaller datasets. These findings highlight that while multi-GPU split-learning effectively reduces training time for large-scale datasets, it may introduce diminishing or even negative returns for smaller hypergraphs with limited computational load. A detailed theoretical analysis of the scalability characteristics of the split-parallel rel-HNN is provided in Appendix G. Additional experimental results on synthetic datasets are provided in Appendix F.

## 6 Conclusions and Future Work

In this paper, we presented rel-HNN, a novel hypergraph neural network framework for learning on relational databases. By representing attribute-value pairs as nodes and tuples as hyperedges, our model captures intricate, fine-grained relationships within and across tuples without relying on schema-specific constraints like primary key–foreign key (PK–FK) relationships. Rel-HNN introduces a multi-level embedding strategy to learn representations at the attribute, tuple, and table levels, offering a comprehensive and expressive approach to relational data modeling. To address the scalability challenge posed by large hypergraphs, we further proposed a split-parallel learning algorithm that effectively distributes the workload across multiple GPUs. Our empirical evaluation shows that rel-HNN consistently outperforms state-of-the-art methods in predictive performance and offers significant computational speedups through parallel training. Building on this foundation, federated hypergraph learning and learning on relational data in cloud environments can be promising directions for future research.

## Acknowledgment

This research was funded by a grant from the Bangladesh Bureau of Educational Information and Statistics (BANBEIS) under Project ID IC20232933. The authors gratefully acknowledge this support.

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

## A    Relational Database and Graph Terminology Mapping

To facilitate the transformation of relational databases into graph or hypergraph structures, we summarize the correspondence used in the literature between relational concepts and their graph counterparts in Table 3. This mapping highlights how relational tuples, attributes, and keys are interpreted as nodes, features, and edges, respectively. Such a representation serves as the foundation for applying graph learning methods to relational data.

Table 3: Relational Database and Graph Terminology Mapping

| Relational Database | Graph Representation |
|---|---|
| Row/Tuple | Node |
| Table | Node type |
| Foreign key column | Edge type |
| Non-foreign-key column | Node features |
| Foreign key from $T_u^A$ to $T_v^B$ | Edge from node $u$ to node $v$ |
| Label of target tuple $t \in T^{tg}$ | Label of node for $t$ of type $tg$ |

## B    Notations

In Table 4, we summarize the notations for node, hyperedge, and table embeddings, as well as the linear and non-linear components of the MLPs employed in both node and hyperedge update steps.

Table 4: Notations for embeddings and transformation functions in rel-HNN.

| Symbol | Description |
|---|---|
| $X_v$ | Input feature vector of node $v \in V$ |
| $Z_v^{(l)}$ | Embedding of node $v$ at layer $l$ |
| $F_e^{(l)}$ | Intermediate embedding of hyperedge $e$ at layer $l$ |
| $F_{e,i}^{(l)}$ | Local partial hyperedge embedding of $e$ computed on GPU $i$ |
| $Z_e^{(l)}$ | Final hyperedge embedding of $e$ at layer $l$ after non-linear activation |
| $Z_T$ | Learnable embedding of table $T$ |
| $\text{MLP}_V^{(l)}$ | Multilayer perceptron applied to nodes at layer $l$ |
| $\text{MLP}_E^{(l)}$ | Multilayer perceptron applied to hyperedges at layer $l$ |
| $\text{MLP}_{E_\mathcal{L}}^{(l)}$ | Linear component of the hyperedge MLP |
| $\text{MLP}_{E_\sigma}^{(l)}$ | Non-linear component of the hyperedge MLP applied after aggregation |

## C    Detailed Algorithms

Algorithm 1 illustrates the procedure for transforming a relational database into a hypergraph representation. Each unique attribute–value pair is mapped to a node, and hyperedges are formed to capture the relationships across tuples within tables. The resulting hypergraph $H = (V, E, X)$ serves as the foundation for the proposed rel-HNN model.

---

**Algorithm 1:** Hypergraph Generation

**Input** : $RDB$: A relational database
**Output:** $H = (V, E, X)$: A hypergraph

**1 begin**
**2**    $V, E, \texttt{node\_map} \leftarrow \emptyset, \emptyset, \{\}$ ;
**3**    **foreach** *table $T^k \in RDB$* **do**
**4**      **foreach** *row $i$ in $T^k$* **do**
**5**        **foreach** *column $j$ in $T^k$* **do**
**6**          **if** $(Attr_j^k, T_{i,j}^k) \notin \texttt{node\_map}$ **then**
**7**            $v \leftarrow new\_node()$ ;
**8**            $\texttt{node\_map}[(Attr_j^k, T_{i,j}^k)] \leftarrow v$ ;
**9**            $V \leftarrow V \cup \{v\}$ ;
**10**    **foreach** *table $T^k \in \mathcal{R}$* **do**
**11**      **foreach** *row $i$ in $T^k$* **do**
**12**        $e \leftarrow \emptyset$ ;
**13**        **foreach** *column $j$ in $T^k$* **do**
**14**          $v \leftarrow \texttt{node\_map}[(Attr_j^k, T_{i,j}^k)]$ ;
**15**          $e \leftarrow e \cup \{v\}$ ;
**16**        $E \leftarrow E \cup \{e\}$ ;
**17**    **foreach** $v \in V$ **do**
**18**      $X[v] \leftarrow$ Feature vector of node $v \in V$ ;
**19**    **return** $H = (V, E, X)$

---

**Algorithm 2:** rel-HNN

**Input** : $H = (V, E, X)$: A hypergraph, $RDB$: A relational database, *epochs*: Number of epochs
**Output:** $\bigcup_{l=0}^{L} \{MLP_V^l \cup MLP_E^l\}$: The $MLP$ parameters for nodes and hyperedges, $\bigcup_{T \in RDB} Z_T$: The table embeddings

**1 begin**
**2**    Initialize parameters $\bigcup_{l=0}^{L} \{MLP_V^l \cup MLP_E^l\}$ and $\bigcup_{T \in RDB} Z_T$ ;
**3**    **for** *epoch $\leftarrow 1$* **to** *epochs* **do**
**4**      **for** $v \in V$ **do**
**5**        $Z_v^0 \leftarrow MLP_V^0(X_v)$;
**6**      **for** $e \in E$ **do**
**7**        $F_e^0 \leftarrow MLP_E^0 \left(\sum_{v \in e} Z_v^0\right)$;
**8**        $Z_e^0 \leftarrow \text{CONCAT}(F_e^0, Z_{T_e})$;
**9**      **for** $l \leftarrow 1 \rightarrow L$ **do**
**10**        **for** $v \in V$ **do**
**11**          $Z_v^l \leftarrow MLP_V^l \left(\sum_{e \in \mathcal{E}_v} Z_e^{l-1}\right)$;
**12**        **for** $e \in E$ **do**
**13**          $Z_e^l \leftarrow MLP_E^l \left(\sum_{v \in e} Z_v^l\right)$;
**14**      $\mathcal{L} \leftarrow loss\_function(\cup_{e \in E} Z_e^L, RDB)$;
**15**      Update the parameters of $\bigcup_{l=1}^{L} MLP_V^l$, $\bigcup_{l=1}^{L} MLP_E^l$ and $\bigcup_{T \in RDB} Z_T$ to minimize $\mathcal{L}$;

---

---

**Algorithm 3:** Split-Parallel rel-HNN

---

**Input** : $H_i = (V_i, E_i, X_i)$: A hypergraph, $RDB$: A relational database, *epochs*: Number of epochs

**Output:** $\bigcup_{l=0}^{L}\{MLP_V^l \cup MLP_{E_\mathcal{L}}^l \cup MLP_{E_\sigma}^l\}$: The $MLP$ parameters for nodes and hyperedges,
$\qquad\bigcup_{T \in RDB} Z_T$: The table embeddings

**1 begin**

**2** $\quad$ Initialize parameters $\bigcup_{l=0}^{L}\{MLP_V^l \cup MLP_{E_\mathcal{L}}^l \cup MLP_{E_\sigma}^l\}$ and $\bigcup_{T \in RDB} Z_T$ ;

**3** $\quad$ **for** *epoch* $\leftarrow 1$ **to** *epochs* **do**

**4** $\qquad$ **for** $v \in V_i$ **do**

**5** $\qquad\quad$ $Z_v^0 \leftarrow MLP_V^0(X_v)$;

**6** $\qquad$ **for** $e \in E_i$ **do**

**7** $\qquad\quad$ $F_{e_i}^0 \leftarrow MLP_{E_\mathcal{L}}^0\left(\sum_{v \in e} Z_v^0\right)$;

**8** $\qquad\quad$ Send $F_{e_i}^0$ to $\cup_{j=1, j \neq i}^N GPU_j$;

**9** $\qquad$ Wait for $\cup_{e \in E_i} \cup_{j=1, j \neq i}^N F_{e_j}^0$ to arrive ;

**10** $\qquad$ **for** $e \in E_i$ **do**

**11** $\qquad\quad$ $F_e^0 \leftarrow MLP_{E_\sigma}^0\left(\sum_{i=0}^N F_{e_i}^0\right)$;

**12** $\qquad\quad$ $Z_e^0 \leftarrow \text{CONCAT}(F_e^0, Z_{T_e})$;

**13** $\qquad$ **for** $l \leftarrow 1 \rightarrow L$ **do**

**14** $\qquad\quad$ **for** $v \in V_i$ **do**

**15** $\qquad\qquad$ $Z_v^l \leftarrow MLP_V^l\left(\sum_{e \in \mathcal{E}_v} Z_e^{l-1}\right)$;

**16** $\qquad\quad$ **for** $e \in E_i$ **do**

**17** $\qquad\qquad$ $F_{e_i}^l \leftarrow MLP_{E_\mathcal{L}}^l\left(\sum_{v \in e} Z_v^l\right)$ ;

**18** $\qquad\qquad$ Send $F_{e_i}^l$ to $\cup_{j=1, j \neq i}^N FPU_j$ ;

**19** $\qquad\quad$ Wait for $\cup_{e \in E_i} \cup_{j=1, j \neq i}^N F_{e_j}^l$ to arrive ;

**20** $\qquad\quad$ **for** $e \in E_i$ **do**

**21** $\qquad\qquad$ $Z_e^l \leftarrow MLP_{E_\sigma}^l\left(\sum_{i=0}^N F_{e_i}^l\right)$;

**22** $\qquad$ $\mathcal{L} \leftarrow loss\_function(\cup_{e \in E} Z_e^L, RDB)$;

**23** $\qquad$ Update the parameters of $\bigcup_{l=1}^{L} MLP_V^l$ , $\bigcup_{l=0}^{L}\{MLP_{E_\mathcal{L}}^l \cup MLP_{E_\sigma}^l\}$, and $\bigcup_{T \in RDB} Z_T$ to
$\qquad$ minimize $\mathcal{L}$;

---

Algorithm 2 presents the pseudocode for the rel-HNN algorithm described in Section 4.2. The algorithm takes the hypergraph $H = (V, E, X)$, the relational database $RDB$, and the epoch number as input and provides model parameters after training as output. The algorithm first initializes the learnable parameters $\bigcup_{l=0}^{L}\{MLP_V^l \cup MLP_E^l\}$ both for the nodes and the hyperedges, respectively. In addition, for each table $T \in$ RDB, we initialize an embedding, $Z_T$, which are learnable parameters of the model (Algorithm 2, Line 2). After determining the class probabilities for the hyperedges corresponding to the tuples in the target table as discussed (Algorithm 2, Lines 4-13), the parameters are updated using backpropagation based on the loss determined previously (Algorithm 2, Line 15).

Algorithm 3 outlines the split-parallel training strategy for rel-HNN as described in Section 4.3. Each GPU processes a shard of the hypergraph and computes local node and edge embeddings. Intermediate edge messages are exchanged across GPUs at every layer, ensuring that dependencies spanning multiple partitions are captured. Node and edge embeddings are then updated iteratively for $L$ layers, and model parameters are optimized by minimizing the loss over the final edge embeddings. This strategy enables efficient multi-GPU training while preserving the integrity of relational dependencies.

## D  Justification of the MLP Decomposition

**Proposition.** Let the hyperedge MLP at layer 0 be defined as

$$\text{MLP}_E^0(x) = \sigma_L\big(W_L \sigma_{L-1}(\cdots \sigma_1(W_1 x + b_1)\cdots) + b_L\big),$$

where each $W_\ell$ and $b_\ell$ denote the weight and bias of layer $\ell$, and $\sigma_\ell(\cdot)$ introduces nonlinearity. We decompose $\text{MLP}_E^0$ into two sequential parts:

$$\text{MLP}_{E_{\mathcal{L}}}^0(x) = W_1 x, \quad \text{MLP}_{E_\sigma}^0(x) = \sigma_L\big(W_L \sigma_{L-1}(\cdots \sigma_2(W_2(\sigma_1(x + b_1)) + b_2)\cdots) + b_L\big).$$

**Claim.** If the first layer is linear and subsequent transformations depend only on the aggregated pre-activation, then computing

$$u = \sum_{i=1}^N \text{MLP}_{E_{\mathcal{L}}}^0(x_i) \quad \text{and} \quad f = \text{MLP}_{E_\sigma}^0(u)$$

is algebraically equivalent to evaluating the full $\text{MLP}_E^0$ on the aggregated input $x = \sum_{i=1}^N x_i$.

**Proof.** Because $W_1$ is linear,

$$\sum_{i=1}^N \text{MLP}_{E_{\mathcal{L}}}^0(x_i) = \sum_{i=1}^N W_1 x_i = W_1 \left( \sum_{i=1}^N x_i \right) = W_1 x.$$

Adding the bias $b_1$ once after aggregation yields the same pre-activation $W_1 x + b_1$ as in the original network. Since all subsequent layers of $\text{MLP}_{E_\sigma}^0$ are applied only to this aggregated activation, their outputs are identical to those of $\text{MLP}_E^0$. Hence,

$$\text{MLP}_{E_\sigma}^0\Big( \sum_i \text{MLP}_{E_{\mathcal{L}}}^0(x_i) \Big) = \text{MLP}_E^0\Big( \sum_i x_i \Big),$$

which proves equivalence. $\square$

# E  Datasets

For the classification task, nine different datasets are selected from various domains: Hepatitis B disease (Hepa), Diabetes disease (Pima), citation networks (Cora), Kinship information (SameGen), Movies information (IMDB), Mutagenicity information (Mutag), Formula 1 dataset (rel-f1(top3) & rel-f1(dnf)), and Online Advertisement (rel-avito(uv)). The first six datasets are from the CTU Relational Learning Repository Motl & Schulte (2024), and the last three datasets are from The Relational Deep Learning Benchmark Repository Robinson et al. (2024). These databases are widely used for evaluating supervised learning models on classification tasks involving relational data, where a categorical attribute is to be predicted. The datasets are selected to reflect a wide range of scales, schema complexities, and classification challenges. The datasets vary from 3 to 10 relational tables and contain up to 70 columns. All datasets are designed for binary classification tasks, except for the Cora dataset, which contains 7 distinct classes. Smaller datasets such as SameGen and Pima provide settings with limited nodes and hyperedges, enabling evaluation of model behavior in low-data. Hepa and MUTAG correspond to moderate-sized datasets with richer relational structures. In contrast, IMDB and rel-avito (uv) represent large-scale real-world datasets with millions of nodes and hyperedges, posing significant challenges in terms of scalability, memory efficiency, and long-range dependency modeling. The rel-f1 (top3) and rel-f1 (dnf) datasets incorporate complex relational schemas with ten tables and high-dimensional feature spaces, enabling rigorous evaluation of model performance on fine-grained, real-world classification tasks under realistic relational settings. The statistics for the datasets used in classification experiments are presented in Table 5.

Table 5: Statistics of Classification Dataset

| Statistic | Hepa | Pima | Cora | SameGen | IMDB | Mutag | rel-f1 (top3) | rel-f1 (dnf) | rel-avito (uv) |
|---|---|---|---|---|---|---|---|---|---|
| # of Nodes | 6488 | 1773 | 7927 | 141 | 4025855 | 13492 | 85263 | 85607 | 3210239 |
| # of Hyperedges | 12927 | 6912 | 57353 | 1536 | 5793251 | 10324 | 76730 | 86742 | 20679117 |
| Total # of Tables | 7 | 9 | 3 | 4 | 7 | 3 | 10 | 10 | 8 |
| Total # of Rows | 12927 | 6912 | 57353 | 1536 | 5793251 | 10324 | 76730 | 86742 | 20679117 |
| Total # of Columns | 26 | 18 | 6 | 8 | 21 | 14 | 70 | 70 | 43 |
| # of Classes | 2 | 2 | 7 | 2 | 2 | 2 | 2 | 2 | 2 |

Table 6: Statistics of Regression Datasets

| Statistic | Pyrimidine | ClassicModels | Pubs | Biodegradability | rel-f1 (position) |
|---|---|---|---|---|---|
| # of Nodes | 130 | 5215 | 681 | 15435 | 85449 |
| # of Hyperedges | 296 | 3864 | 245 | 21895 | 82775 |
| Total # of Tables | 2 | 8 | 10 | 5 | 10 |
| Total # of Rows | 296 | 3864 | 245 | 21895 | 82775 |
| Total # of Columns | 13 | 59 | 61 | 14 | 70 |

Table 6 presents the statistical summary of four relational datasets used for regression tasks in Section 5.3. The first four datasets are from the CTU Relational Learning Repository Motl & Schulte (2024), and the dataset is from The Relational Deep Learning Benchmark Repository Robinson et al. (2024). The datasets exhibit significant variation in relational complexity and scale, enabling a comprehensive evaluation of model robustness and generalization. For instance, *Biodegradability* is the largest in terms of both nodes (15,435) and hyperedges (21,895), suggesting a rich relational structure and potentially complex learning dynamics. In contrast, *Pyrimidine* is the smallest, with only 130 nodes and 296 hyperedges, allowing us to assess performance in low-data regimes. The *ClassicModels* dataset shows a high number of tables (8), columns (59), and classes (273), indicating a detailed schema and a fine-grained prediction task. Similarly, the *Pubs* dataset, despite having a relatively small number of rows (245), has a large number of columns (61), which pose challenges related to feature sparsity or redundancy. Finally, the rel-f1 (position) dataset represents a large-scale real-world regression setting with 85,449 nodes, 82,775 hyperedges, and 10 interconnected tables, enabling evaluation of model scalability and its ability to capture complex, long-range relations.

Table 7 presents the structural statistics of the datasets: *Citeseer*, *DBLP*, and *Pubmed*. Among them, *DBLP* is the largest in terms of both the number of nodes (41,302) and hyperedges (22,363), indicating a highly complex and densely connected hypergraph structure. *Pubmed* represents a medium-sized dataset with 19,717 nodes and 7,963 hyperedges, whereas *Citeseer* is the smallest, comprising 3,312 nodes and 1,079 hyperedges. Despite its relatively small scale, *Citeseer* has the highest feature dimensionality, with each node represented by a 3,703-dimensional feature vector. In contrast, *Pubmed* and *DBLP* exhibit more moderate feature lengths of 500 and 1,425, respectively. The diversity across these datasets in terms of size and feature complexity makes them well-suited for evaluating the effectiveness of the proposed split-parallel hypergraph learning approach.

Table 7: Statistics of Benchmark Hypergraph Datasets

| Property | DBLP | Pubmed | Citeseer |
|---|---|---|---|
| Number of Nodes | 41,302 | 19,717 | 3,312 |
| Number of Hyperedges | 22,363 | 7,963 | 1,079 |
| Length of Feature Vector | 1,425 | 500 | 3,703 |

## F  Split-Parallel Hypergraph Learning Performance on Synthetic Datasets

In addition to real-life benchmark hypergraphs (Section 5.4), we have also experimented with synthetic hypergraphs generated by varying the number of nodes and hyperedges. For each hyperedge, nodes are drawn from a uniform distribution, where the number of nodes varies from three to ten. The feature vectors and labels are also randomly assigned to each node, where the feature vector length is set to 1024. We varied the number of nodes, $|V|$, between 5,000 and 10,000, and the number of hyperedges, $|E|$, between 10,000 and 100,000 to assess performance across different hypergraph scales.

In Figure 5, we present the per-epoch training time across the synthetic hypergraphs for different numbers of GPUs. As the size of the hypergraphs increases—either in terms of the number of nodes or hyperedges—the advantages of parallel learning become more pronounced. For the smallest hypergraph with $|V|$=5,000 and $|E|$=10,000, increasing the number of GPUs leads to an increase in training time, due to synchronization overheads. However, as the number of hyperedges is increased to 50,000 and 100,000, the algorithm achieves

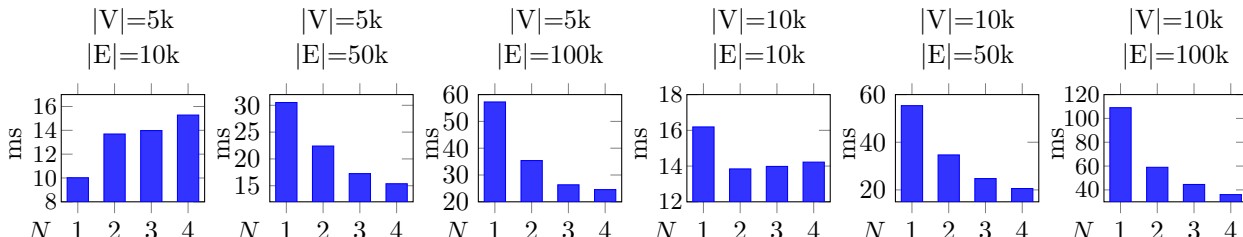

Figure 5: Training time per epoch (in milliseconds) across synthetic hypergraphs for different numbers of GPUs ($N$)

an increasing speedup of $1.98\times$ and $2.33\times$, respectively. For the hypergraph with $|V| = 10{,}000$ and $|E| = 10{,}000$, the training time decreases from 16.19 ms (with a single GPU) to 13.84 ms when using two GPUs. Interestingly, as more GPUs are employed, the training time begins to increase, reaching 14.22 ms with $1.13\times$ speedup for four GPUs, indicating underutilization of parallel resources. Again, increasing the number of hyperedges to 50,000 and 100,000 results in higher speedups of $2.69\times$ and $3.02\times$, respectively. A similar trend is observed with an increasing number of nodes. For instance, with $|E| = 100{,}000$, increasing the number of nodes from 5,000 to 10,000 enhances the speedup from $2.33\times$ to $3.02\times$. These results demonstrate that while synchronization overheads can limit speedup gains on smaller hypergraphs, the proposed parallel learning approach achieves substantial performance improvements as the size and complexity of the hypergraphs scale.

## G    Scalability Analysis of Split-Parallel Hypergraph Learning

For rel-HNN algorithm, the time complexity of computing the initial node embedding $Z_v^{(0)}$ for a node $v \in V$, obtained by applying $MLP_V^{(0)}$ to its input feature vector $X_v$ (Algorithm 2, line 5), is $O(nfd)$, where $n = |V|$ is the number of nodes, $f$ is the input feature dimension, and $d$ is the embedding dimension. For each hyperedge $e \in E$, the computation of the intermediate hyperedge embedding $F_e^{(0)}$ at the initial layer (Algorithm 2, line 7) has a time complexity of $O(end+edd)$, where $e = |E|$ denotes the number of hyperedges. At each subsequent $L$ layers, the complexity of updating the node embedding $Z_v^{(l)}$ (Algorithm 2, line 11) is $O(ned+ndd)$, while determining the hyperedge embedding $Z_e^{(l)}$ requires $O(end+edd)$. The total complexity of a single propagation can be expressed as

$$O\big(L(ndd + end + edd)\big).$$

For the split-parallel rel-HNN executed on two GPUs, the overall time complexity can be expressed as

$$O\big(L\left(\tfrac{1}{2}ndd + \tfrac{1}{2}end + edd + T(ed)\right)\big),$$

where $n$ and $e$ denote the numbers of nodes and hyperedges, respectively, and $L$ is the number of layers. Since the node set is evenly partitioned across two GPUs, the local computation cost for node and hyperedge updates is effectively halved. The term $T(ed)$ represents the communication overhead incurred during inter-GPU synchronization, specifically for exchanging and aggregating local hyperedge embeddings via the AllReduce operation. This communication cost can be modeled as $T(ed) = A + Ced$, where $A$ denotes the latency or startup cost associated with initiating the communication, and $C$ is the per-hyperedge-dimension transfer and aggregation cost that depends on the interconnect bandwidth and the efficiency of the collective communication backend. The split-parallel execution yields a performance advantage when the computational cost of the single-GPU setting exceeds that of the distributed formulation, i.e.,

$$L(ndd + end + edd) \; > \; L\left(\tfrac{1}{2}ndd + \tfrac{1}{2}end + edd + T(ed)\right), \tag{8}$$

where the first term represents the total computation cost on a single GPU and the second term includes the reduced per-GPU computation together with the inter-GPU communication overhead $T(ed)$. This condition

can be simplified as

$$ndd + end \ > \ \tfrac{1}{2}ndd + \tfrac{1}{2}end + T(ed).$$

Substituting the communication model $T(ed) = A + Ced$ into the inequality yields

$$\tfrac{1}{2}ndd + \tfrac{1}{2}end \ > \ A + Ced.$$

Rearranging terms, we obtain

$$\tfrac{1}{2}ndd + ed\left(\tfrac{n}{2} - C\right) \ > \ A,$$

which can be further simplified to

$$e\left(\tfrac{n}{2} - C\right) \ > \ \tfrac{A}{d} - \tfrac{nd}{2}. \tag{9}$$

When the number of nodes $n$ is sufficiently large compared to the communication constants $A$ and $C$, we have $\frac{n}{2} - C > 0$. Using Equation 9, the inequality can be rearranged as

$$e \ > \ \frac{\frac{A}{d} - \frac{nd}{2}}{\frac{n}{2} - C}. \tag{10}$$

As $n$ becomes sufficiently larger than the communication constants $A$ and $C$, the right-hand side of Equation 10 becomes negative. Since the number of hyperedges $e$ is always positive, the inequality is naturally satisfied. Therefore, the split-parallel rel-HNN yields better performance when the number of nodes is significantly large, as the computational gain from parallelization outweighs the communication overhead. Note that, in environments with lower communication overheads (smaller values of $A$ and $C$), the minimum number of nodes for speedup correspondingly decreases.

When the number of nodes $n$ is relatively small compared to the communication constants $A$ and $C$, the term $\frac{n}{2} - C$ becomes negative. In this case, Equation 9 can be rearranged as

$$e \ < \ \frac{\frac{A}{d} - \frac{nd}{2}}{\frac{n}{2} - C}. \tag{11}$$

As the right-hand side of Equation 11 is negative, satisfying the inequality requires the number of hyperedges $e$ to be negative, which is infeasible. This condition indicates that, for smaller graphs or limited node counts, the communication overhead dominates the computation cost, and the split-parallel execution provides no performance gain. In other words, when $n$ is relatively small, the latency and bandwidth terms ($A$ and $C$) outweigh the computational savings from parallelization, making the single-GPU configuration more efficient. The analytical findings are supported by the experimental results. As shown in Figure 4, for the datasets *Pima*, *SameGen*, and *Citeseer*, the single-GPU execution outperforms the split-parallel configuration. From Tables 5 and 7, it can be observed that all these datasets contain relatively few nodes (less than 5,000), which aligns with the analytical conclusion that split-parallel execution is less effective when the number of nodes is small.

## H    Hyperparameter Sensitivity Analysis

To analyze the sensitivity of our model to key hyperparameters, we conducted experiments to evaluate its performance under different parameter settings. Figure 6 presents the AUROC scores of rel-HNN for varying numbers of layers ($L$). In most cases, the best performance is achieved with a shallow architecture ($L = 2$), while deeper configurations ($L > 4$) lead to gradual degradation in accuracy. This trend indicates that rel-HNN effectively captures essential higher-order relationships within only a few message-passing steps, and increasing the depth beyond that introduces redundancy and over-smoothing. For smaller and moderately sized datasets (e.g., *Hepa*, *Pima*, *SameGen*, *Mutag*, and *Cora*), additional layers tend to diffuse information excessively, causing embeddings of connected nodes and hyperedges to become indistinguishable, a well-known phenomenon in graph learning models. Larger datasets, such as *IMDB*, *rel-f1(top3)*, *rel-f1(dnf)*, and *rel-avito(uv)*, exhibit a similar but less pronounced decline, suggesting that even in complex relational structures, shallow architectures are sufficient to achieve optimal relational aggregation. Overall, these results

demonstrate that rel-HNN attains peak performance with two layers, striking a balance between expressive power and generalization while avoiding over-smoothing and unnecessary computation.

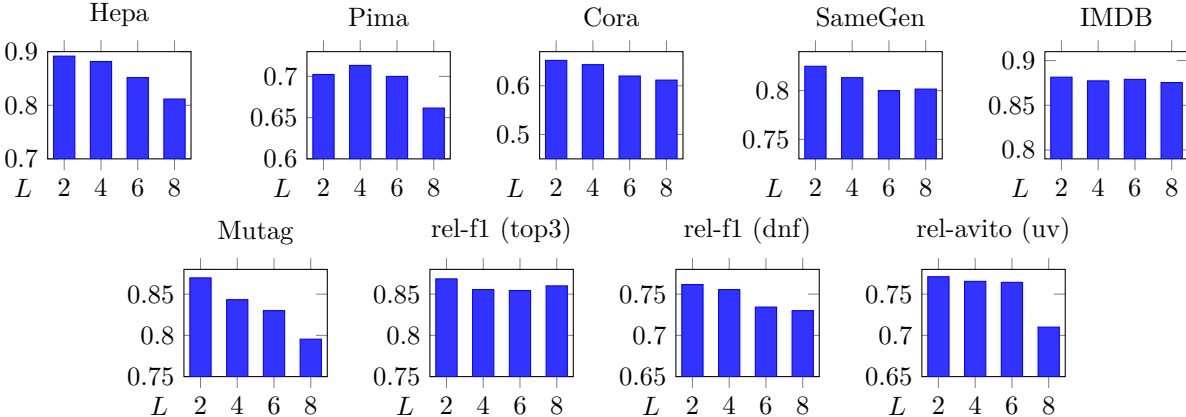

Figure 6: Performance (AUROC) variation with respect to the number of layers ($L$) across datasets.

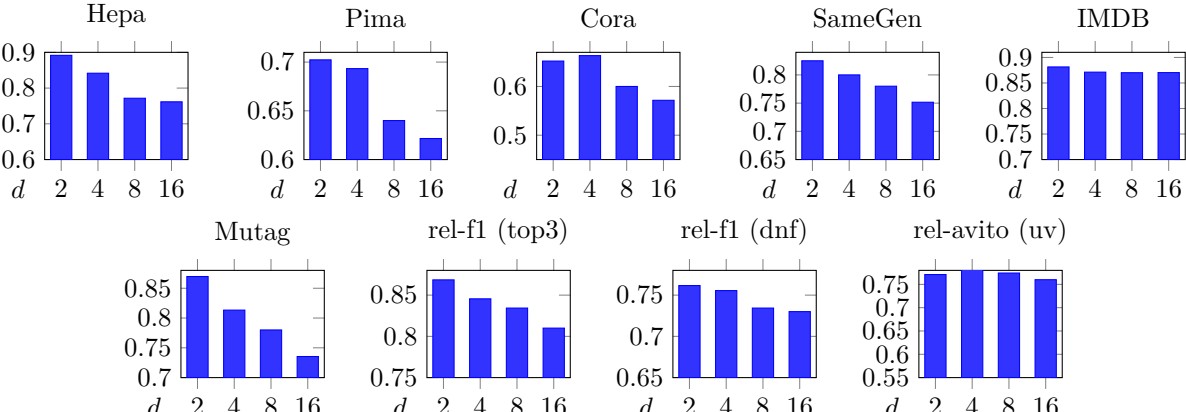

Figure 7: Performance (AUROC) variation with respect to the embedding dimension ($d$) across datasets.

To further analyze the sensitivity of rel-HNN to key hyperparameters, we evaluated its performance across different embedding dimensions ($d$), as shown in Figure 7. In most datasets, the highest AUROC scores are achieved with low-dimensional embeddings ($d = 2$), while performance gradually decreases as the embedding dimension increases. This trend suggests that small embedding spaces are sufficient to capture the underlying relational structure in the data, whereas higher dimensions may introduce redundant or noisy features that hinder generalization. For smaller and moderately sized datasets (e.g., *Hepa*, *Pima*, *Mutag*, and *SameGen*), increasing the embedding dimension leads to overparameterization relative to the available data, resulting in reduced discriminative power. Even for larger relational datasets such as *rel-f1* and *rel-avito*, the improvement from higher-dimensional embeddings remains marginal, with performance saturating or slightly decreasing beyond $d = 4$. Overall, these results indicate that compact embedding representations are sufficient for rel-HNN, providing an efficient trade-off between model complexity, generalization, and training cost.

## I    Memory Consumption Analysis

To analyze the memory characteristics of our model, we conducted experiments using the split-parallel rel-HNN configuration. Figure 8 illustrates the peak GPU memory consumption of the split-parallel rel-

HNN with varying numbers of GPUs ($N$). Across all datasets, a consistent downward trend is observed as $N$ increases, confirming the scalability of our parallelization strategy. As the number of GPUs increases from one to four, memory consumption per GPU decreases substantially — for instance, in the *rel-f1 (top3)* dataset, peak memory usage drops from 5172 MB to 1555 MB, representing a reduction of nearly 70%. However, the decline is not strictly linear, as minor deviations are introduced by the additional communication buffers used during the AllReduce operations required for inter-GPU synchronization. The reduction rate is lower for smaller datasets, such as *SameGen* and *Pima*. Overall, the results validate that the split-parallel rel-HNN achieves effective memory scaling across multiple GPUs, enabling training on larger datasets and higher-dimensional embeddings.

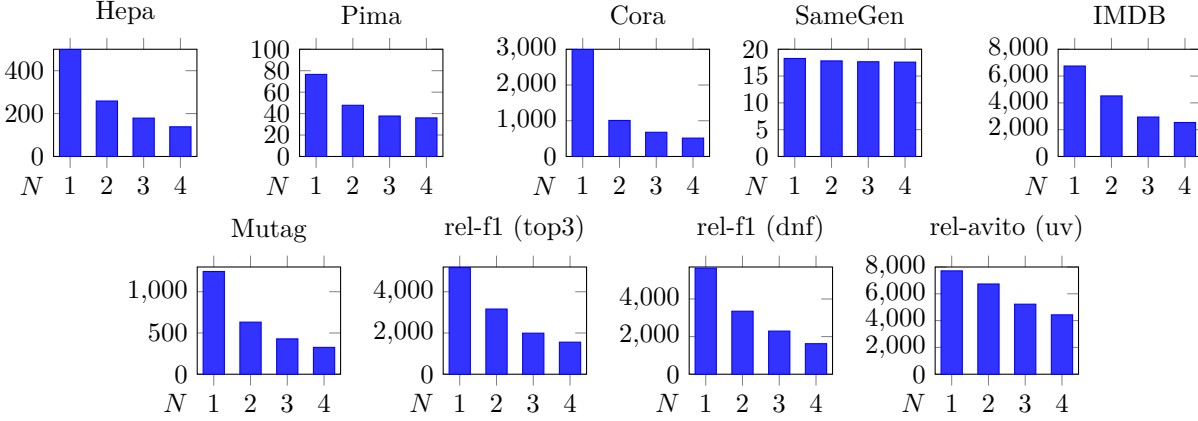

Figure 8: Peak per-GPU memory consumption (MB) with varying numbers of GPUs ($N$) across datasets.