# OpenReview forum: "Rel-HNN: Split Parallel Hypergraph Neural Network for Learning on Relational Databases"
_TMLR — Accepted by TMLR_

### Review · Reviewer_qMXB · 2025-09-10

**Summary Of Contributions:**

This paper introduces Rel-HNN, a hypergraph neural network approach for learning on relational databases that represents attribute-value pairs as nodes and tuples as hyperedges. The key contributions include: (1) a hypergraph representation that captures fine-grained intra-tuple relationships without relying on primary key-foreign key constraints, (2) a multi-level embedding strategy across attribute-value, tuple, and table levels, and (3) a split-parallel GPU training algorithm for scalability on large databases.

**Additional Comments:**

Some figures (especially Figure 3) are cluttered and hard to interpret. Algorithm descriptions could be more precise and writing need to be more precise in places.

**Audience:**

Yes

**Audience Explanation:**

The work applies neural networks to relational databases. This is relevant to the TMLR audience for several reasons:
- Relational databases are ubiquitous in real-world applications
- The hypergraph perspective is practical for structured data representation

**Broader Impact Concerns:**

The paper does not require a broader impact statement. The work focuses on improving machine learning techniques for database analysis, which has generally positive implications for data science applications. No obvious negative societal implications are apparent, though the authors could briefly discuss potential privacy considerations when analyzing sensitive relational data.

**Claims And Evidence:**

No

**Claims Explanation:**

1. The paper primarily compares against older GNN methods (GCN, GAT), SPARE, and ATJ-Net. Critical baselines [1] [2] and recent advances (e.g., DPHGNN [3]) in relational learning and hypergraph neural networks are not adequately covered.
2. Many datasets are relatively small (e.g., Bupa with 495 nodes, SameGebn with 141 nodes), which may not fully validate scalability claims for "large-scale" relational databases.
3. The paper lacks thorough ablation studies to understand which components contribute most to performance gains.

References:
[1] Hypergraph Neural Networks, AAAI 2019
[2] HyperGCN: A New Method For Training Graph Convolutional Networks on Hypergraphs, NeurIPS 2019
[3] DPHGNN: A Dual Perspective Hypergraph Neural Networks, KDD 2024

**Requested Changes:**

1. **Stronger baseline comparison**: Include classic and more recent hypergraph neural network methods (HyperGCN, HGNN, HGNN+) and more recent relational learning approaches as listed in references.
2. **Critical re-evaluation of split-parallel training claims**: The paper claims "substantial speedups" but Figure 4 reveals that split-parallel training only provides benefits on the largest datasets (rel-f1 datasets and DBLP), while actually degrading performance on smaller datasets due to communication overhead. This limits the practical applicability of the approach. There are some suggestions:
   - Clearly specify the dataset size thresholds where split-parallel training becomes beneficial
   - Provide theoretical or empirical guidelines for when to use single-GPU vs multi-GPU training
3. **Better scalability validation**: Test on truly large-scale databases (millions of tuples) to validate scalability claims. Current largest dataset has ~87K tuples.
4. **Ablation studies**: Systematically evaluate the contribution of table embeddings, different node feature encodings, and the multi-level architecture.
5. **Hyperparameter sensitivity**: Analyze sensitivity to key hyperparameters like number of layers, embedding dimensions.
6. **Memory analysis**: Provide detailed memory consumption analysis for the split-parallel approach.

---

> ### Author Response · Authors · 2025-10-16
> **Response to Requested Change #1**
>
> In this reply, we have addressed the specific requested changes regarding stronger baseline comparison and indicated the modified/enhanced portions (highlighted in blue) in the revised manuscript according to the invaluable comments from the reviewer. We would like to express our deep gratitude to the anonymous reviewer for their invaluable comments on the paper. We believe that the constructive suggestions have significantly enriched the quality of the revised paper.
>
> **Response:** We are thankful to the reviewer for his valuable suggestion. In the revision, we have expanded our experimental comparison to include additional strong baselines from both hypergraph neural network and relational learning domains. Specifically, we have incorporated HyperGCN, HGNN, and DPHGNN as representative hypergraph neural network models, covering both classical and recent architectures. Moreover, we have also added RelGT, a graph transformer architecture developed specifically for relational databases, to benchmark against a recent transformer-based relational learning approach. These additions complement our existing relational baselines and provide a more comprehensive evaluation framework. As shown in Table 1, rel-HNN consistently outperforms these models across datasets, demonstrating the effectiveness of the proposed model. We appreciate the reviewer’s insightful suggestions, which have helped us strengthen the experimental depth and completeness of our comparative analysis.

---

> ### Author Response · Authors · 2025-10-16
> **Response to Requested Change #2**
>
> In this reply, we have addressed the specific requested changes regarding critical re-evaluation of split-parallel training claims and indicated the modified/enhanced portions (highlighted in blue) in the revised manuscript according to the invaluable comments from the reviewer. We would like to express our deep gratitude to the anonymous reviewer for their invaluable comments on the paper. We believe that the constructive suggestions have significantly enriched the quality of the revised paper.
>
> **Response:** We appreciate the reviewer’s valuable suggestions. To address this, Appendix G in the revised manuscript includes a comprehensive scalability analysis of split-parallel rel-HNN that quantifies the trade-off between computational parallelization and communication overhead. In addition, the analysis shows that the effectiveness of split-parallel execution primarily depends on the number of nodes. According to the analysis, performance improvements should be observed when the number of nodes is sufficiently large compared to the communication transfer constants. In contrast, for smaller datasets in terms of the number of nodes, single-GPU execution should be more efficient.
>
> This analytical insight is further supported by experimental evidence. As presented in Figure 4, smaller datasets such as Pima, SameGen, and Citeseer (each with fewer than 5,000 nodes) show better performance on a single GPU, whereas datasets with more than 5,000 nodes achieve significant speedups under the split-parallel configuration. Furthermore, in environments with lower communication overhead (smaller A and C), this threshold is expected to decrease as shown in Appendix G, making parallel execution beneficial even for moderately sized datasets.
>
> Together, these results provide both theoretical and empirical justification for the applicability of split-parallel rel-HNN and deliver guidance for determining when split-parallel training offers measurable benefits. We appreciate the reviewer’s constructive feedback, which led to a more rigorous and transparent analysis of the proposed split-parallel training method.

---

> ### Author Response · Authors · 2025-10-16
> **Response to Requested Change #3**
>
> In this reply, we have addressed the specific requested changes regarding better scalability validation and indicated the modified/enhanced portions (highlighted in blue) in the revised manuscript according to the invaluable comments from the reviewer. We would like to express our deep gratitude to the anonymous reviewer for their invaluable comments on the paper. We believe that the constructive suggestions have significantly enriched the quality of the revised paper.
>
> **Response:** We thank the reviewer for his valuable comment regarding the validation of scalability on large-scale databases. In the revised manuscript, we have included an additional large-scale real-world dataset, rel-avito, which contains over 20 million tuples to evaluate the scalability and performance of the proposed rel-HNN model. As shown in Table 1, rel-HNN achieves an AUROC of 0.7712, significantly outperforming the best existing model RelGT(AUROC 0.6874) on this dataset. Figure 4 demonstrates that rel-HNN maintains strong scalability on this dataset, achieving a 3.13× training speedup when increasing the number of GPUs from one to four, while preserving predictive performance. This substantiates the scalability and practical applicability of the proposed split-parallel learning framework for large relational databases. We again appreciate the reviewer’s thoughtful comment, which encouraged us to extend our validation to truly large-scale data, thereby strengthening the empirical robustness of the paper.

---

> ### Author Response · Authors · 2025-10-16
> **Response to Requested Change #4**
>
> In this reply, we have addressed the specific requested changes regarding ablation studies and indicated the modified/enhanced portions (highlighted in blue) in the revised manuscript according to the invaluable comments from the reviewer. We would like to express our deep gratitude to the anonymous reviewer for their invaluable comments on the paper. We believe that the constructive suggestions have significantly enriched the quality of the revised paper.
>
> **Response:** We thank the reviewer for his insightful suggestion. The contribution of table embeddings, node feature encodings, and the multi-level architecture has been systematically evaluated through the rel-HNN variants included in our experiments (Table 1). Specifically, we examined four model versions: rel-HNN-one uses one-hot encoding, while rel-HNN-av employs attribute–value encoding for node features. In both rel-HNN-one and rel-HNN-av, table embeddings are omitted. In contrast, rel-HNN-one-t and rel-HNN-av-t incorporate learnable table embeddings within their respective architectures. As shown in Table 1, the variants with learnable table embeddings and one-hot encoding outperform their counterparts on most datasets and achieve comparable performance on the remaining ones. Additionally, we have added three hypergraph neural network baselines, HyperGCN, HGNN, and DPHGNN, to provide a broader comparison across architectural choices. As rel-HNN consistently outperforms these models in terms of AUROC, the results validate that the proposed two-phase message-passing architecture effectively captures relational dependencies and enhances predictive performance. We appreciate the reviewer’s insightful suggestion, which led to an enhanced demonstration of the architectural component’s contribution.

---

> ### Author Response · Authors · 2025-10-16
> **Response to Requested Change #5**
>
> In this reply, we have addressed the specific requested changes regarding hyperparameter sensitivity and indicated the modified/enhanced portions (highlighted in blue) in the revised manuscript according to the invaluable comments from the reviewer. We would like to express our deep gratitude to the anonymous reviewer for their invaluable comments on the paper. We believe that the constructive suggestions have significantly enriched the quality of the revised paper.
>
> **Response:** We thank the reviewer for his valuable suggestion. In the revised manuscript, we have added a comprehensive Hyperparameter Sensitivity Analysis in Appendix H, where we evaluate the impact of key hyperparameters such as the number of layers (L) and embedding dimension (d) on model performance. The results, presented in Figures 6 and 7, show that rel-HNN achieves peak performance with shallow architectures (L=2) and small embedding dimensions (d=2–4), while deeper or higher-dimensional configurations lead to marginal or negative performance gains due to over-smoothing and overparameterization, a common phenomenon in graph-based learning models. We appreciate the reviewer’s suggestion, which helped us present a more thorough and interpretable analysis of the model’s characteristics with respect to hyperparameter variations.

---

> ### Author Response · Authors · 2025-10-16
> **Response to Requested Change #6**
>
> In this reply, we have addressed the specific requested changes regarding memory analysis and indicated the modified/enhanced portions (highlighted in blue) in the revised manuscript according to the invaluable comments from the reviewer. We would like to express our deep gratitude to the anonymous reviewer for their invaluable comments on the paper. We believe that the constructive suggestions have significantly enriched the quality of the revised paper.
>
> **Response:** We thank the reviewer for his insightful suggestion. In the revised manuscript, we have added a detailed memory consumption analysis for the split-parallel rel-HNN in Appendix I (Figure 8). This analysis reports the peak per-GPU memory usage for varying numbers of GPUs across all datasets. The results show that memory consumption per GPU decreases substantially as the number of GPUs increases, confirming the scalability of our parallelization strategy. We appreciate the reviewer’s thoughtful suggestion, which prompted us to include this detailed analysis and clearly demonstrate the memory scalability of the proposed split-parallel approach.

---

> ### Author Response · Authors · 2025-10-16
> **Response to Additional Comments**
>
> We thank the reviewer for the valuable feedback. In the revised manuscript, we have reorganized the paper to improve readability and logical flow. Specifically, Section 4.3 has been substantially refined to enhance clarity with additional explanations. To remove interpretability, we have also revised several symbols, and a comprehensive summary of the notations is now included in Table 4 (Appendix B). We appreciate the reviewer’s insightful comment, which helped us improve the clarity and presentation of the paper.

---

### Review · Reviewer_DbVS · 2025-09-11

**Summary Of Contributions:**

This paper proposes  a hypergraph neural network framework for learning directly on relational databases(rel-HNN). Instead of flattening data or modeling each tuple as a single node , this model represents each attribute–value pair as a node and each tuple as a hyperedge, thereby capturing fine-grained intra-tuple and inter-tuple relationships.
The model learns multi-level embeddings at the attribute, tuple, and table levels, and introduces a split-parallel training strategy that partitions computation across multiple GPUs to address scalability challenges. Some results across diverse classification and regression benchmarks show consistent improvements over state-of-the-art baselines, alongside significant training speedups.

Pros:
- This is an interesting problem; as data are increasingly stored in databases, a method of this kind could substantially improve the applicability of GNNs in industrial settings.
- The rel-HNN’s embedding design is impressive, as it offers a clear path toward capturing both localized and global relational patterns, thereby enabling richer relational learning
- The experimental results are promising, particularly with respect to the training speedups achieved

Cons:
- Some related work seems to be missing from the discussion, such as recent advances in “Graph Neural Networks for Tabular Data Learning” (CT Li et al., 2024) and hypergraph modeling approaches for databases highlighted in “Graph Neural Networks for Databases: A Survey (Z Li et al.)”. These should be discussed in the paper to provide a more complete context.
- The organization of the paper could be improved; for example, the definition of hypergraphs in Section 3.1 might be more appropriately placed in Section 2？
- Some technical choices are not clearly justified; for instance, it is unclear why Rel-HNN adopts a two-phase message-passing mechanism rather than exploring more phases？

**Audience:**

Yes

**Audience Explanation:**

The paper addresses relational database learning with a hypergraph-based approach, which is of clear interest to TMLR readers in graph learning and data-driven applications

**Broader Impact Concerns:**

The work is primarily methodological and does not present obvious ethical risks. However, since relational databases often contain sensitive data (like medical or financial records), the authors may consider adding a brief note on potential concerns regarding privacy and fairness when deploying such models.

**Claims And Evidence:**

Yes

**Claims Explanation:**

The claims are generally supported by solid experiments and clear comparisons, though certain technical choices and related work discussions could be elaborated further to strengthen the evidence. The code seems can work.

**Requested Changes:**

- **Related Work:** The paper would benefit from a more comprehensive discussion of related literature. In particular, recent advances in *Graph Neural Networks for Tabular Data Learning* (CT Li et al., 2024) and hypergraph modeling approaches for databases discussed in *Graph Neural Networks for Databases: A Survey* (Z Li et al.,  2025) should be included to better situate the proposed method within the existing body of work. This is critical to securing my recommendation.

- **Paper Organization:** The organization of the paper could be improved. For example, the definition of hypergraphs currently presented in Section 3.1 would fit more naturally in Section 2 (Background), which would improve clarity and flow.

- **Technical Justification:** Some design choices are not sufficiently explained. For instance, the rationale for adopting a two-phase message-passing mechanism in Rel-HNN is unclear—further justification for why more phases were not explored would strengthen the contribution.

- **Practical Evaluation:** If the experiments included validation on a real relational database system (e.g., PostgreSQL), it would greatly enhance the usefulness of the paper, since the core motivation is to operate effectively on relational databases.

---

> ### Author Response · Authors · 2025-10-16
> **Responses to the Requested Changes**
>
> In this reply, we have addressed the specific requested changes and indicated the modified/enhanced portions (highlighted in blue) in the revised manuscript according to the invaluable comments from the reviewer. We would like to express our deep gratitude to the anonymous reviewer for their invaluable comments on the paper. We believe that the constructive suggestions have significantly enriched the quality of the revised paper.
>
> **Response to Requested Change #1:** We thank and appreciate the reviewer’s insightful suggestion, which helped us broaden the scope and contextual depth of the related work section, thereby enhancing the positioning of our contributions in the revised version. In the revised manuscript, we have substantially expanded Section 3.1 to include a more comprehensive discussion of recent advances in graph neural networks for both tabular data learning and database-oriented applications. Specifically, we now discuss recent developments in Graph Neural Networks for Tabular Data Learning, covering both common formulations, where instances are represented as nodes and where features are represented as nodes. We have also incorporated recent hypergraph-based tabular learning methods, such as HYTREL. For GNN applications in relational database systems, we have included graph-based methods for performance prediction, query optimization, and cardinality estimation. These additions help situate rel-HNN more clearly within the broader research landscape of graph and hypergraph-based learning for structured data.
>
> **Response to Requested Change #2:** We thank the reviewer for his helpful suggestion. In the revised manuscript, we have reorganized the paper to improve readability and logical flow. The formal definition of hypergraphs has been moved from Section 3.1 to Section 2 (Background), where it now appears alongside other foundational concepts such as relational data representation preliminaries. We appreciate the reviewer’s thoughtful comment, which helped us improve the structural coherence and overall readability of the paper.
>
> **Response to Requested Change #3:** We are grateful to the reviewer and appreciate the reviewer’s insightful comment. In the revised manuscript, we have added a detailed justification in Section 4.2 explaining the rationale behind the two-phase message-passing mechanism in rel-HNN. The two phases: (i) node-to-hyperedge aggregation and (ii) hyperedge-to-node propagation, correspond directly to the underlying structure of hypergraphs. This mechanism ensures full information exchange between nodes and hyperedges within each layer while capturing the structural information. In the context of rel-HNN, these two phases naturally capture the semantics of relational joins: the first phase aggregates attribute-level information within each relation, while the second phase propagates the relational context back to the corresponding attribute-value pairs. To add more phases, additional propagation steps are introduced by stacking multiple layers of rel-HNN. Empirically, we also observed diminishing performance gains for a higher number of phases (Appendix H, Figure 6). These findings support the adoption of a two-phase scheme as an optimal trade-off between model expressiveness and efficiency. We appreciate the reviewer’s detailed observation, which led us to clarify the design rationale.
>
> **Response to Requested Change #4:** We thank the reviewer for his valuable suggestion. We agree that validating the proposed rel-HNN on a real relational database management system (RDBMS) such as PostgreSQL would further strengthen the practical significance of our work. Following the suggestion, we have developed an end-to-end pipeline that directly connects PostgreSQL to the rel-HNN training framework, reproducing the same experimental results and demonstrating how relational data can be efficiently extracted, transformed into hypergraph representations, and used for model training. A brief description of this integration has been added to Section 5.1 of the revised manuscript. Notably, the integration does not require any complex or costly database operations such as multi-way joins. Rather, it relies only on lightweight SQL queries for feature and relation extraction, making the approach easily deployable in existing RDBMS environments. We appreciate the reviewer’s constructive suggestion, which motivated us to extend our experiments to a real database system and thereby strengthen the practical impact of the paper.

---

### Review · Reviewer_dcwG · 2025-10-07

**Summary Of Contributions:**

This paper introduces Rel-HNN, a novel hypergraph-based framework for learning on relational databases. The core idea is to model unique attribute-value pairs as nodes and database rows (tuples) as hyperedges. This schema-agnostic approach effectively captures fine-grained intra-tuple relationships without relying on predefined primary/foreign key constraints. To address scalability, the authors also propose a split-parallel training algorithm that distributes the computation across multiple GPUs.

**Audience:**

Yes

**Audience Explanation:**

Researchers in Graph Neural Networks (GNNs) and Hypergraph Learning: The paper introduces a novel and effective application of hypergraph neural networks, demonstrating their power on a challenging, structured data domain.

The Machine Learning Systems Community: The split-parallel training algorithm for scaling hypergraph learning across multiple GPUs is a direct contribution to research on scalable and distributed ML.

Practitioners and Researchers Working with Structured Data: Applying deep learning directly to relational databases without costly and lossy "flattening" is a major practical problem. This paper offers a high-performing solution that would be widely appealing.

**Claims And Evidence:**

Yes

**Claims Explanation:**

Performance Superiority: The central claim that Rel-HNN outperforms state-of-the-art methods is strongly backed by extensive experimental results presented in Tables 1 and 2. The performance margins on both classification and regression tasks are significant and consistent across a wide variety of datasets.

Novelty of Representation: The proposed hypergraph representation is clearly explained and illustrated in Section 4.1 and Figure 1, substantiating the claim of a novel approach to modeling relational data.

Scalability and Speedup: The claim of computational speedup via the split-parallel algorithm is convincingly demonstrated for large-scale datasets in Figure 4. The evidence is also transparent in showing the limitations on smaller datasets where overhead can diminish gains, which adds to the credibility of the evaluation.

**Requested Changes:**

1.  Clarify the Formulation for Parallel Learning: Refine the mathematical exposition in Section 4.3. Specifically, provide a clear justification for the MLP decomposition (into linear and non-linear parts), unify the notation for local and global embeddings (e.g., G, F, Z), and specify the mechanism for inter-GPU message passing.

2.  Contextualize the HNN Architecture's Novelty: In Section 4.2, please add a brief comparison of the rel-HNN message-passing scheme to a relevant prior hypergraph network (e.g., AllSet). This will help distinguish the architectural innovations from the highly novel data representation.

3.  Provide Deeper Analysis of Scalability: Expand the discussion in Section 5.4 to include a more rigorous analysis of the trade-off between parallelization benefits and communication overheads, especially for smaller datasets. Offering practical guidance or a heuristic for when to use the split-parallel approach would significantly increase the paper's impact.

---

> ### Author Response · Authors · 2025-10-16
> **Responses to the Requested Changes**
>
> In this reply, we have addressed the specific requested changes and indicated the modified/enhanced portions (highlighted in blue) in the revised manuscript according to the invaluable comments from the reviewer. We would like to express our deep gratitude to the anonymous reviewer for their invaluable comments on the paper. We believe that the constructive suggestions have significantly enriched the quality of the revised paper.
>
> **Response to Requested Change #1:** We thank the reviewer for his valuable comment. In the revised manuscript, we have substantially refined Section 4.3 to enhance the mathematical clarity of the split-parallel formulation. We have described the MLP decomposition process explicitly in Section 4.3. In addition, we have provided a formal proof of its correctness in Appendix D to justify the claim more formally. To remove the notational ambiguity, we have revised the symbols. The notations are summarized in Table 4 (Appendix B). Furthermore, we have clarified the inter-GPU communication mechanism for aggregating partial local embeddings. In Section 4.3 of the revised text, we explain that a global tensor with the same ordering of hyperedges is allocated on every GPU. Each GPU fills its corresponding entries with partial vectors (zeros elsewhere) and performs an AllReduce (sum) operation across GPUs to obtain the global aggregated embeddings. We appreciate the reviewer’s insightful suggestion, which significantly improved the mathematical transparency, readability, and reproducibility of our parallel training approach.
>
>
>
> **Response to Requested Change #2:** We sincerely appreciate the reviewer’s thoughtful suggestion. In the revised manuscript, we have expanded Section 4.2 to include a concise comparison between the message-passing mechanism of rel-HNN and relevant prior hypergraph neural networks, particularly AllSet (Chien et al., 2022). Unlike AllSet, which performs symmetric message passing through set functions that aggregate node features into hyperedge representations and then redistribute them to nodes, rel-HNN introduces a relation-aware, table-conditioned aggregation mechanism. It integrates learnable table embeddings and relation-specific transformations. This enables rel-HNN to jointly capture structural dependencies and relational semantics across heterogeneous hyperedges within relational databases. We appreciate the reviewer’s thoughtful feedback, which helped us better articulate and highlight the architectural novelty.
>
> **Response to Requested Change #3:** We sincerely thank the reviewer for the thoughtful and constructive feedback. In the revised manuscript, we have provided an analysis of the scalability characteristics of the split-parallel rel-HNN. Specifically, we now present a quantitative analysis of the trade-off between computational parallelization and communication overhead in Appendix G. The analysis shows that split-parallel execution yields performance gains when the number of nodes is relatively larger than the communication cost components. Conversely, for a lower number of nodes, where communication latency (A) and per-hyperedge-dimension transfer and aggregation cost (C) are significant relative to the per-layer computation, single-GPU execution remains more efficient. Moreover, we have also contextualized this analytical result with experimental evidence: as shown in Figure 4 of the revised manuscript, smaller datasets such as Pima, SameGen, and Citeseer (with fewer than 5,000 nodes) exhibit higher communication overhead and thus perform better on a single GPU, while larger datasets with over 5,000 nodes demonstrate significant speedup with split-parallel execution. These additions provide both theoretical and empirical justification for the scalability behavior of rel-HNN and offer practical guidance on selecting the most efficient training configuration based on dataset size and system parameters. We appreciate the reviewer’s constructive input, which guided us to strengthen the theoretical depth and practical relevance of the scalability analysis.

---

### Decision · Action_Editor_Yzc8 · 2025-11-19

**Recommendation:** Accept with minor revision

**Additional Comments:**

The paper was reviewed by three expert reviewers. One reviewer initially did not agree that the paper's claims were supported by convincing, and clear evidence, due to the outdated baseline methods, the small dataset sizes and the lack of ablation studies. The reviewers addressed all those concerns in the revision: rel-HNN was compared against more recent methods, a large-scale dataset was included, and several model variants were empirically tested. The reviewers also requested improvements to the related work section, better validation of the scalability claims, a study of the model's sensitivity to hyperparameters, and improvements to the organization and presentation of the paper. Most of these issues were addressed in the revision. Following the authors' response and the revision of the manuscript, one reviewer recommended acceptance of the paper, while the other two reviewers were leaning toward acceptance.

Based on the reviews and the recommendations, the consensus is that the contributions of the paper are worth publication. A "minor revision" is recommended for the paper due to some remaining concerns. I request that the final version of the manuscript carefully addresses the items listed below:

- Correct the claim regarding speedups on the hypergraph datasets.

- Improve the presentation of subsections 4.2 and 4.3.

- Evaluate the model on additional large-scale datasets (beyond rel-avito).

- Clearly explain the choice of datasets, including why regression datasets from RelBench were not included.

- Explain why the proposed rel-HNN model outperforms other hypergraph neural networks such as HyperGCN.

- The manuscript was originally submitted as a "Regular submission (maximum 12 pages)", but it now exceeds 13 pages. Please change the submission type to "Long submission".

**Audience:**

Yes

**Audience Explanation:**

Relational data modeling, neural networks for structured data, and hypergraph learning are all active research areas. Therefore, at least some individuals in TMLR's audience would be interested in the paper's findings.

**Claims And Evidence:**

No

**Claims Explanation:**

The following main claims are made in the submission:

- The paper introduces a hypergraph-based representation of relational data, claiming novelty and the ability to preserve both intra-tuple and inter-tuple relationships. To the best of my knowledge, the representation is novel, and the use of hyperedges indeed allows it to capture both intra-tuple and inter-tuple relationships.

- The paper also introduces Rel-HNN, a hypergraph neural network that the authors claim to be tailored for relational databases. The model is designed to operate on the constructed hypergraphs and is therefore suitable for data extracted from relational databases.

- The paper claims that rel-HNN significantly outperforms state-of-the-art methods on classification and regression tasks across different relational datasets. The experimental results shown in Tables 1 and 2 provide evidence supporting this claim.

- Finally, the paper claims that the split-parallel training framework offers computational benefits, achieving substantial speedups on both large-scale relational datasets and on hypergraph datasets. However, this claim is not entirely accurate. For example, on the Citeseer hypergraph dataset, no speedup is observed. The claim should be corrected accordingly.

---

> ### Author Response · Authors · 2025-12-21
> **Response to Action Editor**
>
> We sincerely thank the Action Editor and the reviewers for accepting our paper. We are grateful for the thoughtful and constructive feedback throughout the review process, which has helped us further improve the quality, clarity, and accuracy of the paper. In the final (camera-ready) manuscript, we have carefully addressed the remaining concerns raised by the Action Editor. The specific responses to the comments can be found below:
>
> **Response to Requested Change #1:** We thank the action editor for his valuable comment. In the revised manuscript, we have clarified the speedup claim to accurately reflect our experimental results as suggested. Please refer to the abstract and introduction sections of the revised manuscript. We appreciate the insightful suggestion, which significantly improved the clarity and accuracy of the manuscript.
>
> **Response to Requested Change #2:** We appreciate the constructive suggestion. In the revised manuscript, we have improved the presentation of Sections 4.2 and 4.3 by enhancing the clarity and organization of the exposition. We believe these changes improve readability and accessibility without altering the technical content.
>
> **Response to Requested Change #3:** We thank the action editor for the valuable suggestion. In the revised manuscript, we have added experiments on the IMDB dataset, which is a large-scale relational dataset containing over 5 million tuples. This additional evaluation further demonstrates the scalability and effectiveness of rel-HNN on large relational data. The corresponding results and analysis have been incorporated into the experimental section. We appreciate the insightful suggestion, which helped strengthen the empirical evaluation and better demonstrate the scalability of our approach.
>
> **Response to Requested Change #4:** We are thankful for the insightful suggestion. In the revised manuscript, we have expanded our regression evaluation by including an additional dataset from RelBench, thereby strengthening the empirical coverage of our study. We have also added a detailed explanation in Appendix E, clarifying the dataset selection criteria. We appreciate the suggestion, which motivated a more comprehensive evaluation and significantly improved the clarity of the experimental section.
>
> **Response to Requested Change #5:** We are thankful for the insightful comment. In the revised manuscript, we have explicitly discussed why the proposed rel-HNN model outperforms other hypergraph neural networks as well as the graph neural networks. The HyperGCN model relies on graph-based approximations of hyperedges, which limits the modeling of higher-order interactions, while HGNN and DPHGNN employ fixed aggregation schemes that ignore varying hyperedge roles. The graph-based methods model tuples as monolithic nodes and primarily rely on primary key–foreign key relationships, which limits their ability to capture fine-grained relational dependencies. We appreciate the comment, which helped improve the clarity and depth of our comparative analysis.
>
> **Response to Requested Change #6:** We thank the Action Editor for pointing this out and confirm that the submission is a long paper consistent with the current manuscript length.